# Investigating the relationship of COVID-19 related stress and media consumption with schizotypy, depression, and anxiety in cross-sectional surveys repeated throughout the pandemic in Germany and the UK

Sarah Daimer[1], Lorenz L Mihatsch[2,3], Sharon AS Neufeld[4], Graham K Murray[4,5], Franziska Knolle[1,4]*

[1]Department of Diagnostic and Interventional Neuroradiology, School of Medicine, Technical University of Munich, Munich, Germany; [2]Department of Anaesthesiology and Intensive Care Medicine, Ludwig-Maximilians-Universität München, Munich, Germany; [3]Institute for Medical Information Processing, Biometry and Epidemiology, Ludwig-Maximilians-Universität München, Munich, Germany; [4]Department of Psychiatry, University of Cambridge, Cambridge, United Kingdom; [5]Cambridgeshire and Peterborough NHS Foundation Trust, Cambridge, United Kingdom

*For correspondence: franziska.knolle@tum.de

Competing interest: The authors declare that no competing interests exist.

## Abstract

**Background:** Studies report a strong impact of the COVID-19 pandemic and related stressors on the mental well-being of the general population. In this paper, we investigated whether COVID-19 related concerns and social adversity affected schizotypal traits, anxiety, and depression using structural equation modelling. In mediation analyses, we furthermore explored whether these associations were mediated by healthy (sleep and physical exercise) or unhealthy behaviours (drug and alcohol consumption, excessive media use).

**Methods:** We assessed schizotypy, depression, and anxiety as well as healthy and unhealthy behaviours and a wide range of sociodemographic scores using online surveys from residents of Germany and the United Kingdom over 1 year during the COVID-19 pandemic. Four independent samples were collected (April/May 2020: N=781, September/October 2020: N=498, January/February 2021: N=544, May 2021: N=486). The degree of schizotypy was measured using the Schizotypal Personality Questionnaire (SPQ), anxiety, and depression symptoms were surveyed with the Symptom Checklist (SCL-27), and healthy and unhealthy behaviours were assessed with the Coronavirus Health Impact Survey (CRISIS). Structural equation models were used to consider the influence of COVID-19 related concerns and social adversity on depressive and anxiety-related symptoms and schizotypal traits in relation to certain healthy (sleep and exercise) and unhealthy behaviours (alcohol and drug consumption, excessive media use).

**Results:** The results revealed that COVID-19 related life concerns were significantly associated with schizotypy in the September/October 2020 and May 2021 surveys, with anxiety in the September/October 2020, January/February 2021, and May 2021 surveys, and with depressive symptoms in all surveys. Social adversity significantly affected the expression of schizotypal traits and depressive and anxiety symptoms in all four surveys. Importantly, we found that excessive media consumption (>4 hr per day) fully mediated the relationship between COVID-19 related life concerns and schizotypal

traits in the January/February 2021 survey. Furthermore, several of the surveys showed that excessive media consumption was associated with increased depressive and anxiety-related symptoms in people burdened by COVID-19 related life.

**Conclusions:** The ongoing uncertainties of the pandemic and the restrictions on social life have a strong impact on mental well-being and especially the expression of schizotypal traits. The negative impact is further boosted by excessive media consumption, which is especially critical for people with high schizotypal traits.

**Funding:** FK received funding from the European Union's Horizon 2020 (Grant number 754,462). SN received funding from the Cundill Centre for Child and Youth Depression at the Centre for Addiction and Mental Health, Toronto, Canada and the Wellcome Trust Institutional Strategic Support Fund from the University of Cambridge.

## Editor's evaluation

Using online surveys from Germany and the U.K. participants, this study examined the association between COVID-19 and mental health mediated by participants' behaviors. The results indicate that the pandemic was substantially associated with depressive and anxiety symptoms. Furthermore, unhealthy behaviors such as excessive media consumption further exacerbated the negative impact of social isolation due to COVID.

## Introduction

Starting in Wuhan, China, the novel highly infectious severe acute respiratory syndrome coronavirus 2 (SARS-CoV-2, COVID-19) spread rapidly around the world in the first months of 2020 and was declared as a pandemic by the WHO on 13 March 2020. At the time of writing, more than 258 million people worldwide had contracted the illness, and more than 5.2 million died with or from COVID-19 (*Daly and Robinson, 2021*; *JHU, 2021*). Even after successful vaccination programmes were implemented in many countries (*ECDE, 2021*), the uncertainties and restrictions still impacted many areas of daily life. In many countries, measures such as 'lockdown' or 'social distancing' introduced by governments early on *Cohen and Kupferschmidt, 2020* are now procedures regularly and flexibly applied to combat the pandemic. Many educational institutions, childcare facilities, bars, and restaurants have been closed over long periods during the past 1.5 years, and cultural or sporting events still being cancelled or highly restricted. Governments in different countries differed in their approaches to tackling the pandemic (*Plümper and Neumayer, 2020*). For example, at the start of the pandemic, the United Kingdom (UK) imposed a nationwide lockdown a few days after Germany, which may have led to higher case and mortality rates. However, the UK started vaccinating earlier and faster than Germany, which allowed earlier relaxations of restrictions (*McRobbie et al., 2020*; *Pritchard et al., 2021*). Due to delays in vaccine purchases, significantly less vaccines were available in EU countries, which contributed to fewer vaccinations compared to the UK (*Sasse, 2021*). *Figure 1* shows the number of cases, deaths, and vaccinations per day averaged over the week until June 2021.

Early on, experts expressed concern that the fear of contracting the infection with SARS-CoV-2 as well as pandemic related stressors such as social restrictions, financial preoccupations, task overload, inadequate information, frustration, and boredom posed a risk for mental health, and may be reflected in particular in depressive and anxiety-related symptoms (*Amsalem et al., 2021*; *Brooks et al., 2020*). Social distancing measures in particular represent a macro-stressor affecting the majority of people in an unprecedented manner (*Ahrens et al., 2021*). Different studies indeed showed that severe restrictions of social contacts as well as the fear of the virus or the impact on living conditions had a measurable impact on the mental health of general populations all over the world (*Bu et al., 2020*; *Smith et al., 2020*; *Wang et al., 2020b*). During the first lockdown, increased levels of perceived stress and mental distress, COVID-19 related fear, general anxiety and depression, and a general decline in mental well-being were observed in many countries, including Germany and the UK (*Bäuerle et al., 2020*; *Fancourt et al., 2021*; *Proto and Quintana-Domeque, 2021*; *Smith et al., 2020*).

In this stressful context, there is the additional risk that people engage more in unhealthy behaviour and coping strategies, such as excessive consumption of alcohol or drugs, potentially due to insecurity and social restrictions (*Clay and Parker, 2020*; *Marsden et al., 2020*; *McKay and Asmundson,*

**eLife digest** The 2020 COVID-19 pandemic, and the measures different governments took to contain it, harmed many people's mental well-being. The restrictions, combined with pandemic-related uncertainty, caused many individuals to experience increased stress, depression, and anxiety. Many people turned to unhealthy behaviours to cope, including consuming more alcohol or drugs, using media excessively, developing poor sleeping habits, or reducing the amount of exercise they did.

Stress, drugs, poor sleep, and uncertainty can increase an individual's risk of developing psychotic symptoms, including delusions, hallucinations, or difficulty thinking clearly. These symptoms may be temporary or part of a more lasting condition, like schizophrenia. The risk of developing these symptoms increases in people with 'schizotypal traits', such as a lack of close relationships, paranoia, or unusual or implausible beliefs. These individuals may be especially vulnerable to the harmful mental health effects of the pandemic.

Daimer et al. demonstrated that people who were more worried about their life stability or financial situation during the 2020 COVID-19 pandemic had worse mental well-being than those who felt secure. In the experiments, volunteers completed a series of online mental health questionnaires at four different time points during the pandemic. People who reported feeling lonely, having negative thoughts, or experiencing fewer positive social interactions had more symptoms of mental illness. People who experienced more life disruptions also reported more anxiety or depression symptoms and more schizotypal traits. Daily consumption of at least four hours of digital media exacerbated negative mental health symptoms, and people with more pandemic-related life concerns also spent more time on digital media

Daimer et al. suggest that increased media consumption among people with pandemic-related hardships may have increased mental health symptoms and schizotypal traits in these individuals. The survey results suggest that maintaining a healthy lifestyle, including meaningful relationships, is essential to staying mentally healthy during extreme situations like a global pandemic. Protective interventions – such as strengthening social support networks, providing mental health education, or increasing mental healthcare provisions – are essential to prevent poor mental health outcomes during future crises.

*2020*; *Pfefferbaum and North, 2020*). One of the first signs of this was the rise in supermarket sales of alcohol in the UK in response to pub and restaurant closures. In the week ending 21 March , alcohol sales rose by 67%, but total supermarket sales increased by only 43% (*Finlay and Gilmore, 2020*). However, reports on general alcohol consumption during the pandemic have diverse findings. Some studies reported an increase in alcohol consumption in one fifth to one third of the participants during the first weeks of the COVID-19 pandemic in spring 2020 (*Daly and Robinson, 2021*; *Jacob et al., 2021*; *Taylor et al., 2021*). A study in 21 European countries showed that alcohol consumption generally decreased in all countries except for the UK, where alcohol consumption increased, and Ireland, where consumption remained the same (*Kilian et al., 2021*). However, the literature shows that the patterns of alcohol consumption change in the course of the pandemic, that is, fewer episodes of heavy drinking (*Kilian et al., 2021*), but also that particular groups are at increased risk, for example, individuals with children, and those with higher depressive or anxiety scores (*Sallie et al., 2020*). Furthermore, people who have already had an addiction problem or other psychiatric diagnosis appear particularly at risk. Marsden and colleagues showed that while the normal population did not increase their drinking behaviour, those who already showed harmful or problematic drinking patterns increased their consumption (*Marsden et al., 2020*). The consumption of other substances, including legal, illegal, and prescriptive drugs, may also have increased in response to the pandemic in general (*Manthey et al., 2021*), and specifically to cope with COVID-19 related stress (*Czeisler et al., 2020*). People with a substance use disorder may increase their consumption in reaction to the negative impact of the situation, shift to other substances if access to their primary substances becomes limited, or relapse if they have already recovered from the addiction (*Chiappini et al., 2020*).

Besides substance use, many people may also 'escape' from the current situation through excessive media consumption or use media excessively in search for information regarding the current

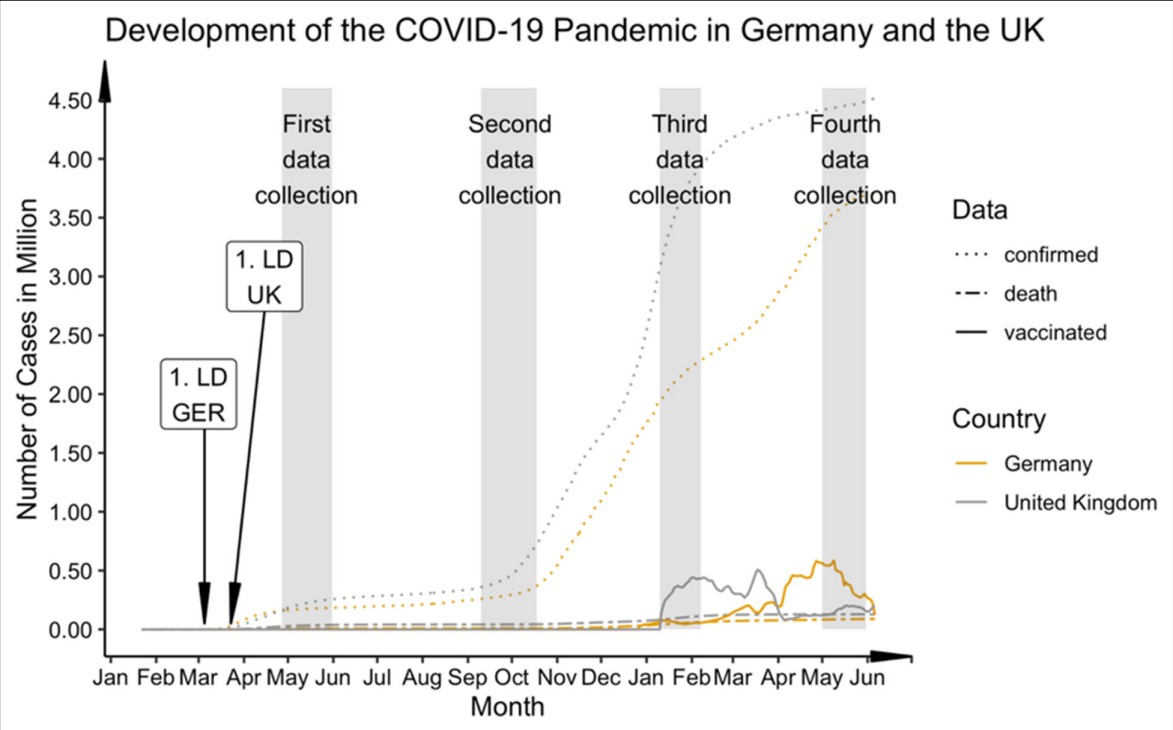

**Figure 1.** Progression of COVID-19 cases, deaths, and vaccine rate (first dose) in comparison between Germany and the UK from January 2020 to June 2021. Germany: 1. State-wise lockdown (LD) on 16 March 2020; UK: 1. National LD on 23 March 2020. Numbers of vaccinations averaged over the week. Data (cases and deaths) taken from the 2019 Novel CoronaVirus CoViD (2019-nCoV) data repository by Johns Hopkins University Center for Systems Science and Engineering (JHU CSSE). Data for UK vaccination rate taken from GOV.UK Coronavirus in the UK (https://coronavirus.data.gov.uk/details/vaccinations) and for the German vaccination rate from Impfdashboard (https://impfdashboard.de/daten). The grey bars mark the time sections of data collection in the present study.

situation. The media plays a particular role in how individuals cope with disasters (*Taylor, 2019*), which also applies to the COVID-19 pandemic. On the one hand, the media provides essential information about the virus, recent developments, and protective measures; further, social media is necessary to stay in contact with others and may also be useful to distract oneself from boredom due to a lack of alternatives (*Bendau et al., 2021*). On the other hand, the media may produce rumours or misinformation (*Tasnim et al., 2020*), or perpetuate the sense of constant threat, which may lead to increased anxiety (*Garfin et al., 2020*; *Satici et al., 2020*). Interestingly, the WHO recently started using the term 'infodemic' in the context of the present crisis as 'a tsunami of misinformation, hate, scapegoating, and scare-mongering (which) has been unleashed' (*WHO, 2020*). A large body of literature suggests that excessive media consumption (>3 hr daily) is associated with poorer mental health (*Abbas et al., 2021*; *Bendau et al., 2021 Gao et al., 2020*; *Neophytou et al., 2019*; *Ni et al., 2020*; *Su et al., 2021*; *Valdez et al., 2020*; *Wang et al., 2020a*; *Zendle, 2020*).

In the context of the COVID-19 pandemic, most general-population studies focus on the increase of symptoms of common mental disorders, that is, depression and anxiety. The present study extends this work by also addressing the development of schizotypal traits. Our previous studies indicated an increase in schizotypic traits during the pandemic (*Daimer et al., 2021*; *Knolle et al., 2021*). Schizotypy describes a multidimensional unifying construct that represents the underlying vulnerability for schizophrenia-spectrum psychopathology that is expressed across a broad range of personality, subclinical, and clinical psychosis phenomenology (*Debbané and Barrantes-Vidal, 2015*; *Grant and Hennig, 2020*; *Raine, 1991*). It is regarded as a latent multidimensional personality trait that occurs in varying degrees in every person along a continuum (*Grant et al., 2018*). There is no consensus on the exact content of the construct, but all definitions include positive (e.g. paranoia), negative (i.e. lack of trust), and disorganised traits, similar to schizophrenia (*Wright et al., 2011*).

The pandemic could pose an increased risk for people with pronounced schizotypal traits. For example, paranoia related to the risk of a contagious virus may result in individuals experiencing

higher levels of psychological stress, depressive feelings, and anxiety (*Preti et al., 2020*). Fekih-Romdhane and colleagues showed that people scoring high on schizotypy were more likely to think they had COVID-19 symptoms than people with low schizotypy, and they also experienced more pronounced fear of illness (*Fekih-Romdhane et al., 2021*). Furthermore, a comparison between individuals with low and high schizotypy during the pandemic showed that those with low schizotypy had a variety of strategies to adaptively cope with the fear of COVID-19 (e.g. positive reframing, planning, religion, self-distraction, and behavioural disengagement), yet those with high schizotypy had fewer such strategies (*Fekih-Romdhane et al., 2021*). Those with high schizotypy were more likely to use maladaptive strategies such as venting or self-blame, which were less effective in reducing COVID-19 related anxiety (*Fekih-Romdhane et al., 2021*). Due to greater levels of introversion and mistrust of others, individuals with high schizotypal traits have fewer and poorer social contacts compared to others, are less integrated into communities, and are more likely to be lonely (*Kozloff et al., 2020*; *Le et al., 2019*). When contact with others is regarded as an additional potential risk to one's health, schizotypal individuals may further reduce their social contacts, which may take longer to be restored in the aftermath of the pandemic (*Preti et al., 2020*). Additionally, individuals with high schizotypal traits may be particularly likely to succumb to media misinformation, for example, due to their low agreeableness (*Kwapil et al., 2018*) and disposition towards believing conspiracist ideation (*Barron et al., 2014*). A recent study found that people with pronounced schizotypy were more likely to share false political information (*Buchanan and Kempley, 2021*). Buchanan and Kempley argued that such individuals are less likely to question the truth of information than those low in schizotypy. This suggests that people scoring high on schizotypy are more likely to become victims of misinformation in the media. Misinformation due to excessive media consumption following mental health deterioration has been frequently reported during this pandemic (*De Coninck et al., 2021*; *Su et al., 2021*; *Tasnim et al., 2020*; *Zhao and Zhou, 2020*; *Zhong et al., 2020*).

People with a high schizotypal personality are at an increased risk of developing a schizophrenic disorder (*Linscott et al., 2018*). According to the diathesis-stress model, environmental factors, such as major psychological stressors or stressful life events, cause the onset of psychosis in the presence of a biological predisposition to schizophrenia (*Debbané and Barrantes-Vidal, 2015*). The psychological stress caused by the prospect of being infected by the COVID-19 virus, as well as the social restraining measures to mitigate the virus' spread, could potentially trigger the development of a full-blown psychosis in people with high schizotypy (*Carter et al., 1995*; *Chapman et al., 1994*; *Grant and Hennig, 2020*). Valdes-Florido and colleagues (2020) reported cases of psychosis triggered by the fear of infection, compulsory home-confinement or concerns about the economic consequences of the lockdown. *Hu et al., 2020* reported a 25% increase in the incidence of schizophrenia in January 2020 compared to previous years in China. The authors attribute this to the psychosocial stress and social distancing measures caused by COVID-19 (*Hu et al., 2020*). These medium and long-term effects of COVID-19 as an adverse life event could disproportionately affect people at risk of psychosis over the next few years (*Beards et al., 2013*; *Brown et al., 2020*).

In this paper, we therefore aim to investigate whether pandemic-related stressors, namely 'COVID-19 related social adversity' and 'COVID-19 related life concerns' have an effect on mental health, and whether these effects are mediated by healthy and unhealthy behaviours. We generated a separate latent model for each stressor across four samples, each collected at a different time point over a year from the start of the pandemic. We hypothesised that the separate COVID-19 stressors related to restrictions and financial concerns ('life concerns'), and inhibited social relationships ('social adversity'), would each be negatively associated with symptoms of common mental distress (i.e. depressive and anxiety-related symptoms), and schizotypal traits, at all time points during the pandemic. In addition, we hypothesised that this association will be mediated by healthy and unhealthy behaviours (respectively, exercise and sleep; alcohol, media, and drug consumption). Specifically, we expect that individuals who experience high levels of the above-mentioned stressors may damage their mental well-being through the unhealthy behaviours of substance use and excessive media consumption, and may improve their mental well-being through the healthy behaviours of adequate sleep and regular exercise.

## Materials and methods

### Study design and procedure

The COVID-19 exposure and COVID-19 related mental health questionnaire was designed as an online survey using EvaSys (https://www.evasys.de, Electric Paper Evaluationssysteme GmbH, Luneburg, Germany), see *Knolle et al., 2021* for a detailed description. The questionnaires were available in German and English. For participant recruitment we used a snowball sampling strategy to reach the general public in Germany and the UK. The April/May 2020 survey was collected from 27 April 2020 to 31 May 2020, approximately 5 weeks after the introduction of the nationwide lockdowns in the UK and Germany. Completion of this first survey took approximately 35 min. The three follow-up surveys took only about 15 min, as they did not include question on subjective change of psychological distress (September/October 2020 survey: 10 September 2020 to 18 October 2020; January/February 2021 survey: 10 January 2021 to 7 february 2021 ; May 2021 survey: 1 May 2021 to 31 May 2021). For each cohort, the recruitment goal was 400 subjects. With this sample size, small effect sizes of $f$=0.02 may be detected with a power of above 80% using linear two-tailed regressions. *Figure 1* shows the time point of data collection with respect to the development of cases and deaths in the UK and Germany. The subjects were contacted at the first time point via print media (i.e. Sueddeutsche Zeitung) and social media (i.e. Facebook, Twitter, and WhatsApp) and asked to forward the questionnaire to friends and family according to the snowball sampling strategy. At the following sampling time points, respondents who had provided their email address were contacted again and additional participants were recruited via social media and recruitment platforms (https://www.call-forparticipants.com). Participation was voluntary, and participants did not receive any compensation.

Ethical approval was obtained from the Ethical Commission Board of the Technical University Munich (250/20 S). All participants provided informed consent.

### Survey measures

The self-report online survey was conducted using three standardised questionnaires: the Symptom Checklist 27, the Schizotypal Personality Questionnaire (SPQ) (*Raine, 1991*), the CRISIS survey, and additional demographic information (age, self-reported gender, education and parental education, and living conditions). The Coronavirus Health Impact Survey (CRISIS, http://www.crisissurvey.org/) (*Nikolaidis et al., 2021*) was used to assessed COVID-19 exposure (infection status, symptoms, contact), subjective mental and physical health, and healthy and unhealthy behaviour (i.e. weekly amount of sleep and exercise, consumption of alcohol, drugs, and media). The survey was created by *Merikangas et al., 2020* in the wake of the COVID-19 crisis in order to enable researchers and care providers to examine the extent and impact of life changes induced by the epidemic on mental health and behaviours of individuals and families across diverse international settings. A recent study using a large pilot sample from the US and UK with the CRISIS found that the results are highly reproducible and indicate a high degree of consistency in the predictive power of these factors for mental health during the COVID-19 pandemic (*Nikolaidis et al., 2021*).We assessed the general mental health status using the short form of the Symptom Check List (SCL-27) (*Hardt et al., 2011*; *Hardt and Gerbershagen, 2001*; *Kuhl et al., 2010*). The SCL's 27 items assess symptoms of anxiety, depression, mistrust, and somatisation. It is a commonly used, economical, multidimensional screening instrument with good psychometric properties (*Hardt et al., 2011*; *Hardt and Braehler, 2007*; *Kuhl et al., 2010*). Finally, we assessed schizotypy using the SPQ(dichotomous version). It is a useful and widely used screening for schizotypal personality disorder in the general population (*Raine, 1991*).

### Variables and statistical analysis

Statistical analysis and visualisations were computed using R and R Studio (*RStudio Team, 2020*). We used Wilcoxon rank sum tests or Chi-square test of independence to explore differences between the countries (UK, Germany) and four surveys (April/May 2020, September/October 2020, January/February 2021, May 2021) on demographics and the COVID-19 exposure variables.

To further explore the differences between the countries and time points in CRISIS variables we conducted robust ANOVAS (*Mair and Wilcox, 2020*) using the R package WRS2 with country (UK, Germany) and survey (April/May 2020, September/October 2020, January/February 2021, May 2021) as a between-subjects factor. This method was chosen because the data did not meet the parametric assumptions of ANOVA (normality, equal variance, outliers) and the robust ANOVA version is

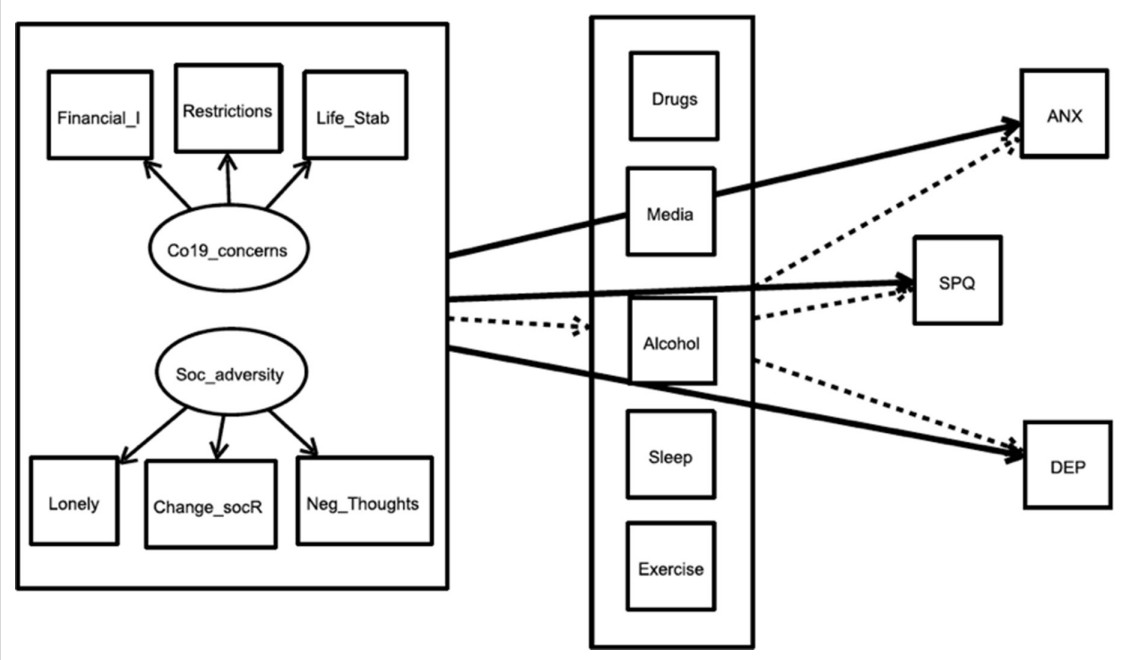

**Figure 2.** Overview of the complete theoretical model for the influence of 'social adversity' and 'COVID-19 related life concerns' directly on mental health scores and indirectly via healthy and harmful behaviour variables. The same model was calculated separately for the two exogenous variables ('predictors', left large box) for all four time points. The bold arrows indicate the direct pathways to the three endogenous variables ('outcomes'). The dashed arrows indicate the indirect pathways from predictor to outcomes via the five mediators (box in the middle). Co19_concerns: COVID-19 related life concerns; Soc_adversity: social adversity; financial_I: financial impact due to the crisis; Restrictions: Perception of the restrictions as stressful; Life_Stability: Concerns about life stability due to the crisis; Change_socR: stressful social relationship changes; Neg_Thoughts: Negative Thoughts during COVID 19; ANX: anxiety symptoms; DEP: depressive Symptoms, SPQ: total sum scores of SPQ.

designed for non-parametric data using a maximum-likelihood estimator and therefore less sensitive to non-normality, unequal variance, and outliers compared to normal ANOVA.

In order to assess the relationships between possible influences of health-damaging behaviour on depressive and anxiety-related symptoms as well as high schizotypy, we created structural equation models (SEM) using the Lavaan package in R (*Rosseel, 2012*). The visualisations of the models were carried out using Onyx (*Brandmaier, 2021*). See *Figure 2* for an overview of the hypothesised models.

Our three endogenous manifest variables ('outcomes') were schizotypy trait, symptoms of depression, and symptoms of anxiety. Schizotypal trait was defined as the total sum score of all SPQ items. For anxiety and depressive symptoms, specific items of the SCL-27 were summed and divided by the number of items. For depression, we used all SCL-27 items assessing dysthymic and depressive symptoms; and for anxiety we used all items describing agoraphobic and social phobic symptoms.

For the exogenous latent variables ('predictors'), we generated two variables. The first variable, which we call 'COVID-19 related life concerns', reflects the direct impact of the pandemic on the individual's life. The best fitting measurement model was formed using the CRISIS items 'to what degree have changes related to the coronavirus/COVID-19 crisis in your area created financial problems for you or your family?', 'how stressful have the restrictions on your daily life been for you?' and 'to what degree are you concerned about the stability of your living situation?'. The exogenous variables were scaled so that we fixed the variance of all factor loadings within one exogenous latent variable to 1. This allowed comparing estimates between different models using these exogenous latent variables quantitively. The second exogenous latent variable, 'social adversity', reflects loneliness and whether change in the quality of social relationships was perceived as stressful. The variables that loaded best on the factor 'social adversity' were the CRISIS items 'has the quality of the relationships between you and members of your family/social contacts changed/ how stressful have these changes in contacts been for you?', 'how lonely were you?' and 'to what extent did you have negative thoughts, thoughts about unpleasant experiences or things that make you feel bad?'. The two measurement models for the two exogenous latent variables (predictors) had excellent fit, see *Supplementary file 1a*.

The mediators drug use and excessive media use were recoded into dichotomous variables from categorical responses, due to sparse endorsement of some categories. Both were expressed with a positive value for at least one occasion of marijuana, opiate or tranquilliser use in the last 4 weeks, or, for at least 4 hr of daily consumption of any of social media, digital media, or video game (none at all, under 1 hr, 1–3 hr, 4–6 hr, more than 6 hr), respectively. Alcohol consumption was used as a continuous variable (not at all, rarely, once a month, several times a month, once a week, several times a week, once a day, more than once a day). Hours of sleep per night (<6 hr, 6–8 hr, 8–10 hr, >10 hr) during the week and frequency of 30 min exercise (sporting activities) per week were analysed with the original categories from the CRISIS measure.

For each of the four survey time points, two separate models were conducted, one to assess how 'social adversity' predicted mental health scores, and the second to assess how 'COVID-19 related life concerns' predicted mental health scores. Mental health scores were our three endogenous variables, SPQ traits, depressive symptoms, and anxiety symptoms. Covariances between the outcome variables were included in the model, including covariances between mediators led to a non-converging model. As we hypothesised that these relationships are mediated by healthy and unhealthy behaviours, we included drug consumption, excessive media use, alcohol consumption, hours of sleep per night during the week, and exercise per week as mediation variables to explain the relationship between each exogenous latent variable ('COVID-19 related life concerns', 'social adversity') and all the endogenous variables (SPQ, depression, and anxiety). In order to facilitate comparison across the four surveys collected at different time points throughout the pandemic, we included as confounders in each model those variables which were associated with differences across the time points at p<0.01 (*Table 1*). We did not control for suspected COVID-19 infection, as the infection rate was <1% in each of the four samples. We also included age and gender as these are known confounders of many of the paths in our proposed model. As each survey consisted of different participants, each model was based on cross-sectional data. Therefore, the direction of the associations may be ambiguous and the variables may influence each other. Nonetheless, here we examine the influence of distressing conditions, which have occurred in particular because of the COVID-19 pandemic, on mental health, and the mediation of these relationships by healthy and unhealthy behaviours. Alternative models are presented in the supplements. We fitted mediation models that included the direct effect of the exogenous latent variable on the endogenous variable (pathway c) as well as the indirect effects via the mediation variables (pathway a+b), which included the effect of the exogenous variable to the mediator (pathway a) and the mediator to the endogenous variable (pathway b). The total effect is calculated as the product of paths a and b plus the direct effect c. *Figure 2* shows an overview of the theoretical models. Note that the models were calculated separately for the two exogenous latent variables and the samples at the different time points.

All models were estimated using maximum likelihood estimation combined with bootstrapped errors. To examine the quality of the model fit, we consulted the Comparative-Fit-Index (CFI; *Bentler, 1990*) for the relative model fit and the root mean squared error (RMSEA; *Steiger, 1990*) for the absolute model fit. An adequate fit is assumed for RMSEA of less than 0.06 and a CFI of greater than 0.90 (*Beauducel and Wittmann, 2005*; *Browne and Cudeck, 1992*).

## Results

### Participants and sample comparisons

Sample descriptions for all four surveys are presented in *Table 1*. The first survey (April/May 2020) was completed by 781 participants (UK N=239; Germany N=542), after excluding three participants who did not provide consent. The second survey (September/October 2020) was completed by 498 participants (Germany, N=383; UK N=115), after excluding three responders under the age of 18, a prerequisite for taking part in the study. In the third survey (January/February 2021), 544 participants were included after the exclusion of one person who did not give consent to the participation and four further participants under the age of 18 (Germany N=448; UK N=96). A total of 486 (Germany N=416; UK N=70) people participated in the last survey, in May 2021, after two were excluded due to lack of consent and two for being under the age of 18.

Since participation at one time point was not required for taking part in another one, we regard all four samples as independent. 26 participants took part in all four surveys. The samples did not

**Table 1.** Cohort demographics and COVID-19 exposure divided by country and time point.
Values in percent if not indicated otherwise.

| | | Survey 1 April/May 2020 | | Survey 2 September/October 2020 | | Survey 3 January/February 2021 | | Survey 4 May 2021 | | Country comparison | | | Survey comparison | | |
|---|---|---|---|---|---|---|---|---|---|---|---|---|---|---|---|
| | | UK | Ger | UK | Ger | UK | Ger | UK | Ger | W/X² | df | p | K/X² | df | p |
| N | | 239 | 542 | 115 | 383 | 96 | 448 | 70 | 416 | | | | | | |
| Percent | | 27.9 | 63.1 | 21.1 | 70.4 | 16.3 | 76.1 | 13.7 | 81.6 | | | | 55.31 | 3 | 0.000 |
| Age | Mean | 39.01 | 45.36 | 40.9 | 42.87 | 42.05 | 43.18 | 44.04 | 43.84 | 398,256 | | 0.000 | 2.85 | 3 | 0.416 |
| | SD | 16.02 | 14.87 | 16.17 | 16.04 | 17.38 | 14.45 | 18.73 | 14.64 | | | | | | |
| Gender | Male | 24.3 | 25.9 | 29.6 | 25.1 | 21.9 | 29.7 | 22.9 | 24.5 | 0.69 | 2 | 0.709 | 6.76 | 6 | 0.343 |
| | Female | 73.6 | 71.2 | 68.7 | 72.8 | 75.0 | 68.5 | 74.3 | 74.8 | | | | | | |
| | other/ n.a. | 2.1 | 3.0 | 1.7 | 2.1 | 3.1 | 1.8 | 2.9 | 0.7 | | | | | | |
| Education | School leavers | 0.4 | 0.0 | 0.0 | 0.0 | 0.0 | 0.0 | 0.0 | 0.2 | 194.74 | 8 | 0.000 | 125.74 | 24 | 0.000 |
| | 8 years | 19.2 | 13.1 | 16.5 | 13.8 | 17.7 | 25.9 | 5.7 | 20.9 | | | | | | |
| | Prof. college | 31.8 | 21.6 | 39.1 | 30.8 | 49.0 | 32.6 | 44.3 | 34.1 | | | | | | |
| | Master or < | 47.3 | 64.9 | 43.5 | 55.1 | 33.0 | 41.3 | 48.6 | 44.7 | | | | | | |
| | Missing | 1.3 | 0.4 | 0.9 | 0.3 | 0.0 | 0.2 | 1.4 | 0.0 | | | | | | |
| Living Area | City | 20.50 | 60.30 | 25.20 | 43.30 | 24.00 | 37.10 | 20.00 | 47.40 | 147.64 | 5 | 0.000 | 52.74 | 15 | 0.000 |
| | Suburb | 7.90 | 13.10 | 12.20 | 13.80 | 14.60 | 11.80 | 14.30 | 10.10 | | | | | | |
| | Town | 36.40 | 10.70 | 26.10 | 14.10 | 15.60 | 18.30 | 35.70 | 14.20 | | | | | | |
| | Village, rural | 34.70 | 15.70 | 35.70 | 26.40 | 44.80 | 31.50 | 30.00 | 26.00 | | | | | | |
| | Missing | 0.40 | 0.20 | 0.90 | 2.30 | 1.00 | 1.30 | 0.00 | 2.40 | | | | | | |
| Rating physical health before Co19 | Exc. | 13.00 | 11.30 | 8.70 | 10.70 | 13.50 | 15.80 | 8.60 | 13.90 | 28.49 | 5 | 0.000 | 29.62 | 15 | 0.013 |
| | Very good | 32.20 | 33.30 | 25.20 | 32.90 | 27.10 | 36.20 | 34.30 | 38.20 | | | | | | |
| | Good | 31.80 | 35.70 | 42.60 | 42.00 | 33.30 | 34.60 | 34.30 | 35.10 | | | | | | |
| | Fairly | 17.20 | 15.50 | 18.30 | 11.0 | 15.60 | 9.60 | 17.10 | 9.10 | | | | | | |
| | Poor | 3.80 | 3.30 | 4.30 | 1.60 | 10.40 | 1.80 | 5.70 | 1.70 | | | | | | |
| | Missing | 2.10 | 0.90 | 0.90 | 1.80 | 0.00 | 2.00 | 0.00 | 1.90 | | | | | | |
| Treatment physical illness | No | 66.90 | 72.80 | 71.30 | 81.50 | 72.90 | 76.10 | 75.70 | 76.20 | 26.27 | 2 | 0.000 | 101.25 | 6 | 0.000 |
| | Yes | 18.40 | 20.10 | 27.00 | 17.80 | 26.0 | 22.80 | 22.90 | 23.10 | | | | | | |
| | Missing | 14.60 | 7.00 | 1.70 | 0.80 | 1.00 | 1.10 | 1.40 | 0.70 | | | | | | |

*Table 1 continued on next page*

*Table 1 continued*

| | | Survey 1 April/May 2020 | | Survey 2 September/October 2020 | | Survey 3 January/February 2021 | | Survey 4 May 2021 | | Country comparison | | | Survey comparison | | |
|---|---|---|---|---|---|---|---|---|---|---|---|---|---|---|---|
| | | UK | Ger | UK | Ger | UK | Ger | UK | Ger | W/X² | df | p | K/X² | df | p |
| | Exc. | 14.20 | 15.30 | 8.70 | 11.50 | 15.60 | 17.90 | 15.70 | 15.10 | 87.86 | 5 | 0.000 | 41.85 | 15 | 0.000 |
| | Very good | 21.80 | 37.00 | 23.50 | 33.40 | 27.10 | 40.80 | 25.70 | 42.10 | | | | | | |
| | Good | 30.10 | 28.30 | 38.30 | 37.90 | 28.10 | 28.30 | 32.90 | 27.20 | | | | | | |
| | Fairly | 21.30 | 14.60 | 18.30 | 12.30 | 16.70 | 9.40 | 14.30 | 11.80 | | | | | | |
| Rating mental health before Co19 | Poor | 11.30 | 3.30 | 9.60 | 3.90 | 10.40 | 2.50 | 11.40 | 2.90 | | | | | | |
| | Missing | 1.30 | 1.50 | 1.70 | 1.00 | 2.10 | 1.10 | 0.00 | 1.00 | | | | | | |
| | No | 69.00 | 81.90 | 79.10 | 89.60 | 78.10 | 86.40 | 80.00 | 87.70 | 41.63 | 2 | 0.000 | 92.40 | 6 | 0.000 |
| Psych. Treatment | Yes | 17.60 | 10.90 | 18.30 | 9.70 | 20.80 | 12.70 | 20.00 | 10.80 | | | | | | |
| | Missing | 13.40 | 7.20 | 2.60 | 0.80 | 1.00 | 0.90 | 0.00 | 1.40 | | | | | | |
| | Positive Test | 0.00 | 0.20 | 0.00 | 0.80 | 5.20 | 3.10 | 1.40 | 3.80 | 41.76 | 12 | 0.004 | 41.76 | 12 | 0.000 |
| | Diagn. | 2.50 | 0.70 | 0.90 | 0.50 | 1.00 | 0.90 | 0.00 | 0.50 | | | | | | |
| | Sympt. | 18.80 | 14.20 | 19.10 | 15.40 | 21.90 | 16.50 | 28.60 | 14.70 | | | | | | |
| | No inf. | 78.70 | 83.90 | 79.10 | 82.50 | 71.90 | 79.20 | 70.00 | 80.80 | | | | | | |
| Suspected infection | Missing | 0.00 | 0.90 | 0.90 | 0.80 | 0.00 | 0.20 | 0.00 | 0.20 | | | | | | |

GER: German sample; UK: UK sample; TP: time point; p: p-vlaue; n.a.: missing values; Exc.: Excellent; Diagn: positive diagnosis; Sympt: Symptoms of COVID-19; No inf: No infection.

differ significantly in terms of age ($X^2$=2.85, p=0.416) and gender ($X^2$=6.76, p=0.343). However, in the second, third, and fourth survey, significantly more participants came from Germany ($X^2$=55.31, p<0.001). In addition, the distributions of educational level ($X^2$=125.74, p<0.001) and living area ($X^2$=52.74, p<0.001) differed significantly between the time points (*Table 1*).

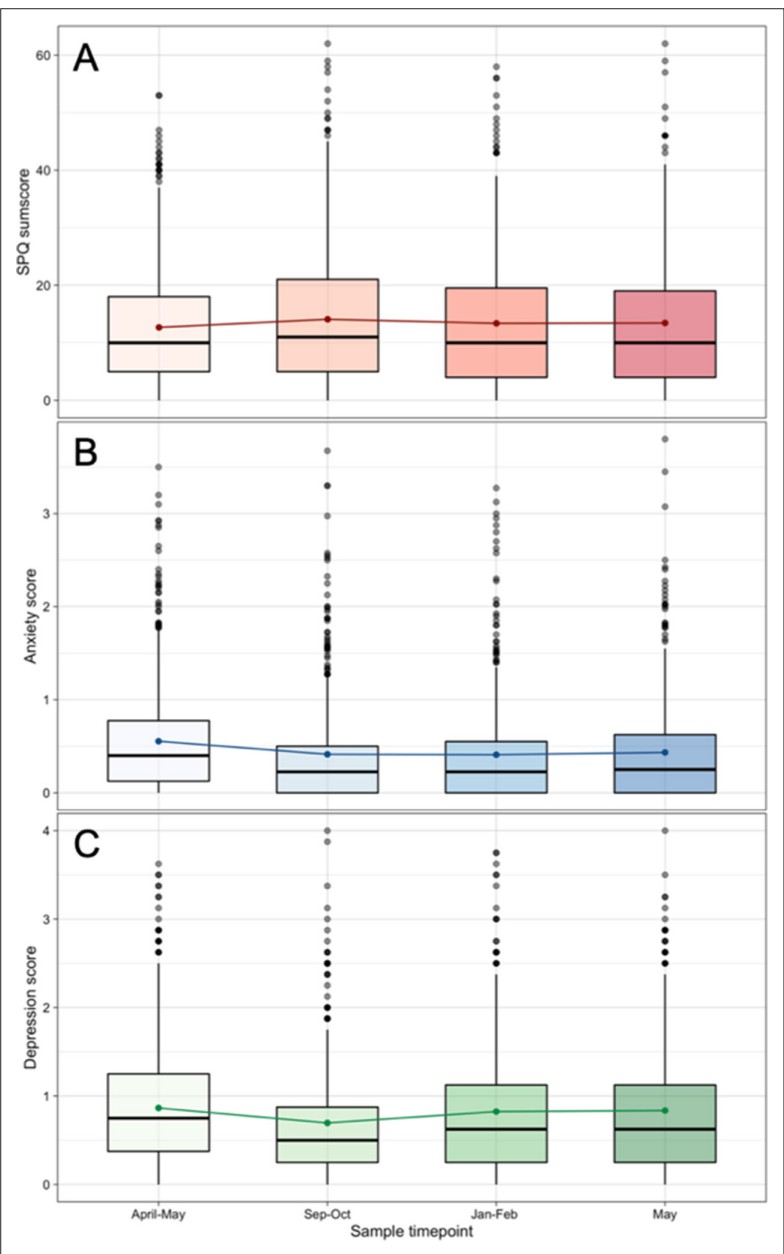

**Figure 3.** Boxplot of (A) Schizotypal Personality Questionnaire (SPQ), (B) anxiety scores (SCL-27) and (C) depression scores (SCL-27) separately for four time point samples. The colored dots represent the means of the distributions, the boxplot the median (middle line in the box), the lower and upper quartiles (edges of the box), the area where 95% of the distribution lies (vertical lines), and the outliers (black dots). The respective sample sizes are April-May sample N = 781, September – October sample N = 498, January – February sample N = 544, May sample N = 486.

**Table 2.** Overview of the results of the Kruskal-Wallis Chi square test and Dunn's post-hoc test for the endogenous variables, mediators, or exogenous latent variables in the following structural equation models.

Comparisons presented are those comparing the latter survey to the prior survey, as well the last survey to the first survey.

| | Survey comparison | | Post-hoc test (Dunn's test p-values) | | | |
|---|---|---|---|---|---|---|
| | Kruskal-Wallis $X^2$ | p | April/May 2020 vs September/ Oct. 2020 | September/ October 2020 vs January/ February 2021 | January /February 2021 vs May 2021 | April/May 2020 vs May 2021 |
| **Endogenous variable** | | | | | | |
| SPQ | 0.61 | 0.893 | 0.58 | 0.44 | 0.79 | 0.99 |
| Anxiety | 57.56 | 0.000 | 0.00 | 1.00 | 0.37 | 0.00 |
| Depression | 28.04 | 0.000 | 0.00 | 0.00 | 0.93 | 0.09 |
| **Mediators** | | | | | | |
| Excessive media use | 39.98 | 0.000 | 0.00 | 0.00 | 0.00 | 0.92 |
| Drug consumption | 1.92 | 0.590 | 0.98 | 0.57 | 0.58 | 0.21 |
| Alcohol consumption | 28.40 | 0.000 | 0.02 | 0.32 | 0.11 | 0.00 |
| Exercise | 78.98 | 0.000 | 0.00 | 0.00 | 0.92 | 0.00 |
| Sleep | 20.96 | 0.000 | 0.01 | 0.43 | 0.58 | 0.00 |
| **Exogenous variables** | | | | | | |
| Restrictions stressful | 33.49 | 0.000 | 0.00 | 0.00 | 0.01 | 0.34 |
| Financial impact | 25.89 | 0.000 | 0.00 | 0.13 | 0.01 | 0.00 |
| Concerns life stability | 35.65 | 0.000 | 0.00 | 0.00 | 0.00 | 0.01 |
| Lonely | 18.13 | 0.000 | 0.00 | 0.00 | 0.56 | 0.28 |
| Negative Thoughts | 13.93 | 0.002 | 0.00 | 0.01 | 0.99 | 0.41 |
| Stressful social relationship changes | 62.25 | 0.000 | 0.00 | 0.00 | 0.68 | 0.89 |

p: significance value; SPQ: Schizotypal Personality Questionnaire.

### Expression of schizotypal traits, depressive, and anxiety-related symptoms in the samples collected at four time points during the COVID-19 pandemic

The Kruskal-Wallis test indicated no difference in SPQ scores across the samples at the four time points, but revealed significant differences in anxiety and depressive symptoms across the samples at the four time points (*Figure 3A–C*; *Table 2*). As a control analysis we conducted generalised estimate equation model (*Carey et al., 2022*). Data from all four surveys were pooled, with the first survey acting as the reference for our three outcome variables; the results match those of the Kruskal-Wallis test (*Supplementary file 1b*). Pearson's correlation coefficient indicated significant moderate correlations between SPQ and anxiety ($r=0.57$–$0.68$) and between SPQ and depression ($r=0.53$–$0.58$) in all four surveys (*Supplementary file 1c*).

### Structural equation modelling

The two models showed an acceptable model fit at all four time points, with a RMSEA of ≤.08, the Tucker-Lewis index (TLI) and CFI ≥0.8. Variables and items relevant for the models are described in *Table 2*, for different samples at different time points. An overview of all model fits is shown in *Table 3*. The models were calculated with all participants. We then performed a sensitivity analysis calculating the same models without the 27 subjects who participated at all four time points. The models fit equivalently to the full sample models and findings remained unchanged (see *Supplementary file 1d*). The full model statistics for the model is presented in *Supplementary file 1e–1l*. The two models which are presented here are highly complex, containing five mediators and three outcomes. Predictor measurement models show close to perfect fit (*Supplementary file 1a*). Reduced models

**Table 3.** Overview of the model fit indices separated by exogenous latent variable and survey.

| | | | | Exact model fit | Relative model fit | |
| | | Test statistic | df | $X^2$ | CFI | RMSEA |
|---|---|---|---|---|---|---|
| **Predictor** | **Survey** | | | | | |
| | 1 - April/May 20 | 480.16 | 135 | 0.000 | 0.836 | 0.073 |
| | 2 - Sept./ Oct. 20 | 440.01 | 135 | 0.000 | 0.862 | 0.072 |
| | 3 - Jan. /Feb. 21 | 426.43 | 135 | 0.000 | 0.883 | 0.066 |
| COVID-19 related life concerns | 4 - May 21 | 413.53 | 135 | 0.000 | 0.867 | 0.068 |
| | 1 - April/May 20 | 492.39 | 135 | 0.000 | 0.853 | 0.074 |
| | 2 - Sept./ Oct. 20 | 481.87 | 135 | 0.000 | 0.857 | 0.078 |
| | 3 - Jan. /Feb. 21 | 498.07 | 135 | 0.000 | 0.866 | 0.074 |
| Social adversity | 4 - May 21 | 448.08 | 135 | 0.000 | 0.866 | 0.073 |

df: degree of freedom; $X^2$: Chi squared test; CFI: comparative fit index; RMSEA: root mean square error of approximation.

only containing one outcome have much improved model fits, with a RMSEA of ≤0.07, the TLI and CFI ≥0.9; see *Supplementary file 1m*.

## 'COVID-19 related life concerns' model

The results for this model are presented in *Table 4*, and for the mediator media in *Figure 4*.

Generally, the structural equation model revealed that the exogenous latent variable 'COVID-19 related life concerns' was directly related to schizotypy (September/October 2020, May 2021), anxiety (September/October 2020, January/February 2021, May 2021), and depression (all surveys) (pathway c), and 'COVID-19 related life concerns' was positively associated with media (January/February 2021), drug (September/October 2020, January/February 2021, May 2021) and alcohol consumption (May 2021), and negatively associated with sleep (January/February 2021, May 2021) (pathway a). There were significant indirect effects of 'COVID-19 related life concerns' on schizotypy, depression, and anxiety only through excessive media consumption (pathway a–b). The alternative models reveal a worse model fit and are presented in *Supplementary file 1n*, for Akaike information criterion and Bayesian information criterion (AIC/BIC )in comparison.

Investigating the endogenous variable SPQ (*Figure 4*, *Table 4*), we found that concerns directly related to the COVID-19 pandemic had an increasing effect on SPQ scores in the September/ October 2020 survey and May 2021 survey, with this estimate being highest at the September/ October 2020 survey (c=0.23). Here, more stress due to the COVID-19 restrictions led to higher SPQ scores. At these time points as well as at the January/ February 2021 survey, the total effects were significant. The mediation of this effect via media consumption was significant in the January/ February 2021 survey. People who experienced greater COVID-19 stress were more likely to consume media excessively and have higher SPQ. Media consumption explained a quarter of the total effects of COVID-19 stress on SPQ In addition, there was a trend towards a significant mediation with sleep in the January/ February 2021 survey.

Investigating anxiety (*Figure 4*, *Table 4*), we found in the last three surveys that COVID-19 related concerns were associated with increased anxiety, with the strongest effect at the May 2021 survey (c=0.20) and the weakest effect in January/ February 2021 (c=0.14). Significant total effects were measurable at these time points. A significant indirect effect with the mediator excessive media consumption was seen in January/ February 2021: 12.5% of the effects of COVID-19 related concerns on anxiety symptoms occurred through excessive media consumption.

In all four surveys, increased stress from the COVID-19 pandemic was significantly related to greater depressive symptoms (*Figure 4*, *Table 4*), with this effect being lowest (c=0.22) at the April/ May 2020 survey and highest (c=0.35) in May 2021. Therefore, all total effects were also significant. Only in January/ February 2021 was there a significant indirect effect with the mediator excessive media consumption: here, excessive media consumption explained 6% of the effects of COVID-19 stress on depressive symptoms. No other statistically significant mediation effects were found.

**Table 4.** Overview of all results of the structural equation model with 'COVID-19 related life concerns' as exogeneous latent variable for mental health endogenous variables mediated by harmful and healthy behaviours.
The results are separated by endogenous variable (schizotypal traits, depressive symptoms, anxiety symptoms), mediators (excessive media use, drug consumption, alcohol consumption, units of exercise per week, hours of sleep per night during weeks) and samples at the four time points, showing indirect, direct, and total effects, as well as individual pathways.

| Outcome | Mediators | Sample time point | Indirect effect | | Total effect | | a | | b | | c | |
|---|---|---|---|---|---|---|---|---|---|---|---|---|
| | | | estimate | p | estimate | p | estimate | p | estimate | p | estimate | p |
| SPQ | Excessive Media use | 1 | 0.01 | 0.181 | 0.01 | 0.915 | 0.05 | 0.078 | 0.23 | .020 | 0.00 | 0.950 |
| | | 2 | 0.00 | 0.201 | 0.23 | 0.001 | 0.01 | 0.828 | 0.26 | 0.011 | 0.23 | 0.001 |
| | | 3 | 0.03 | .015 | 0.12 | .040 | 0.10 | .000 | 0.31 | .001 | 0.09 | .131 |
| | | 4 | 0.00 | 0.839 | 0.20 | 0.001 | 0.04 | 0.196 | 0.02 | 0.799 | 0.20 | 0.001 |
| | Drug consumption | 1 | 0.00 | 0.561 | 0.00 | 0.997 | 0.03 | 0.266 | 0.15 | 0.318 | 0.00 | 0.950 |
| | | 2 | 0.01 | 0.427 | 0.23 | 0.001 | 0.04 | 0.009 | 0.14 | 0.295 | 0.23 | 0.001 |
| | | 3 | 0.10 | 0.430 | 0.10 | 0.077 | 0.09 | 0.000 | 0.11 | 0.400 | 0.09 | 0.131 |
| | | 4 | 0.01 | 0.334 | 0.21 | 0.000 | 0.06 | 0.018 | 0.15 | 0.277 | 0.20 | 0.001 |
| | Alcohol consumption | 1 | 0.00 | 0.761 | 0.00 | 0.974 | 0.20 | 0.271 | 0.01 | 0.625 | 0.00 | 0.950 |
| | | 2 | 0.00 | 0.872 | 0.22 | 0.001 | -0.04 | 0.764 | 0.01 | 0.480 | 0.23 | 0.001 |
| | | 3 | 0.00 | 0.701 | 0.09 | 0.137 | 0.08 | 0.535 | -0.02 | 0.320 | 0.09 | 0.131 |
| | | 4 | -0.01 | 0.226 | 0.19 | 0.001 | 0.33 | 0.017 | -0.03 | 0.136 | 0.20 | 0.001 |
| | Exercise (units per week) | 1 | 0.00 | 0.506 | 0.00 | 0.989 | -0.08 | 0.340 | -0.04 | 0.218 | 0.00 | 0.950 |
| | | 2 | 0.00 | 0.589 | 0.22 | 0.001 | -0.10 | 0.133 | 0.03 | 0.479 | 0.23 | 0.001 |
| | | 3 | 0.00 | 0.871 | 0.09 | 0.133 | -0.11 | 0.059 | 0.01 | 0.852 | 0.09 | 0.131 |
| | | 4 | 0.00 | 0.531 | 0.20 | 0.001 | -0.10 | 0.114 | 0.04 | 0.421 | 0.20 | 0.001 |
| | Sleep (hours per night) | 1 | 0.01 | 0.307 | 0.01 | 0.934 | -0.08 | 0.110 | -0.12 | 0.073 | 0.00 | 0.950 |
| | | 2 | 0.01 | 0.369 | 0.23 | 0.000 | -0.02 | 0.332 | -0.30 | 0.015 | 0.23 | 0.001 |
| | | 3 | 0.02 | 0.081 | 0.11 | 0.042 | -0.17 | 0.000 | -0.15 | 0.057 | 0.09 | 0.131 |
| | | 4 | 0.00 | 0.909 | 0.20 | 0.000 | -0.09 | 0.002 | -0.02 | 0.906 | 0.20 | 0.001 |
| Anxiety | Excessive Media use | 1 | 0.01 | 0.240 | 0.07 | 0.131 | 0.05 | 0.078 | 0.10 | 0.094 | 0.07 | 0.160 |
| | | 2 | 0.00 | 0.844 | 0.18 | 0.000 | 0.01 | 0.828 | 0.11 | 0.052 | 0.18 | 0.000 |
| | | 3 | 0.02 | 0.008 | 0.16 | 0.000 | 0.10 | 0.000 | 0.17 | 0.001 | 0.14 | 0.000 |
| | | 4 | 0.00 | 0.266 | 0.20 | 0.000 | 0.04 | 0.196 | 0.12 | 0.021 | 0.20 | 0.000 |
| | Drug consumption | 1 | 0.00 | 0.726 | 0.07 | 0.147 | 0.03 | 0.266 | 0.05 | 0.591 | 0.07 | 0.160 |
| | | 2 | 0.00 | 0.384 | 0.19 | 0.000 | 0.04 | 0.009 | 0.11 | 0.218 | 0.18 | 0.000 |
| | | 3 | 0.01 | 0.121 | 0.16 | 0.000 | 0.09 | 0.000 | 0.14 | 0.083 | 0.14 | 0.000 |
| | | 4 | 0.00 | 0.951 | 0.20 | 0.000 | 0.06 | 0.018 | -0.01 | 0.965 | 0.20 | 0.000 |

*Table 4 continued on next page*

*Table 4 continued*

| Outcome | Mediators | Sample time point | Indirect effect estimate | p | Total effect estimate | p | a estimate | p | b estimate | p | c estimate | p |
|---|---|---|---|---|---|---|---|---|---|---|---|---|
| | Alcohol consumption | 1 | 0.00 | 0.726 | 0.07 | 0.147 | 0.20 | 0.271 | 0.01 | 0.343 | 0.07 | 0.160 |
| | | 2 | 0.00 | 0.961 | 0.18 | 0.000 | -0.04 | 0.764 | 0.00 | 0.857 | 0.18 | 0.000 |
| | | 3 | 0.00 | 0.736 | 0.14 | 0.000 | 0.08 | 0.535 | -0.01 | 0.443 | 0.14 | 0.000 |
| | | 4 | 0.00 | 0.818 | 0.20 | 0.000 | 0.33 | 0.017 | 0.00 | 0.802 | 0.20 | 0.000 |
| | Exercise (units per week) | 1 | 0.00 | 0.696 | 0.07 | 0.167 | -0.08 | 0.340 | 0.01 | 0.607 | 0.07 | 0.160 |
| | | 2 | 0.00 | 0.540 | 0.18 | 0.000 | -0.10 | 0.133 | 0.02 | 0.018 | 0.18 | 0.000 |
| | | 3 | 0.00 | 0.675 | 0.14 | 0.000 | -0.11 | 0.059 | 0.01 | 0.615 | 0.14 | 0.000 |
| | | 4 | -0.01 | 0.240 | 0.19 | 0.000 | -0.10 | 0.114 | 0.06 | 0.015 | 0.20 | 0.000 |
| | Sleep (hours per night) | 1 | 0.01 | 0.196 | 0.08 | 0.128 | -0.08 | 0.110 | -0.09 | 0.043 | 0.07 | 0.160 |
| | | 2 | 0.00 | 0.425 | 0.19 | 0.000 | -0.02 | 0.332 | -0.13 | 0.073 | 0.18 | 0.000 |
| | | 3 | 0.00 | 0.916 | 0.15 | 0.000 | -0.17 | 0.000 | -0.01 | 0.915 | 0.14 | 0.000 |
| | | 4 | 0.00 | 0.805 | 0.20 | 0.000 | -0.09 | 0.002 | -0.02 | 0.794 | 0.20 | 0.000 |
| Depression | Excessive Media use | 1 | 0.00 | 0.282 | 0.22 | 0.003 | 0.05 | 0.078 | 0.08 | 0.179 | 0.22 | 0.004 |
| | | 2 | 0.00 | 0.851 | 0.28 | 0.000 | 0.01 | 0.828 | 0.10 | 0.128 | 0.28 | 0.000 |
| | | 3 | 0.02 | 0.010 | 0.32 | 0.000 | 0.10 | 0.000 | 0.19 | 0.002 | 0.31 | 0.000 |
| | | 4 | 0.00 | 0.342 | 0.36 | 0.000 | 0.04 | 0.196 | 0.11 | 0.139 | 0.35 | 0.000 |
| | Drug consumption | 1 | 0.00 | 0.603 | 0.22 | 0.004 | 0.03 | 0.266 | 0.08 | 0.439 | 0.22 | 0.004 |
| | | 2 | 0.01 | 0.381 | 0.28 | 0.000 | 0.04 | 0.009 | 0.14 | 0.223 | 0.28 | 0.000 |
| | | 3 | 0.01 | 0.176 | 0.32 | 0.000 | 0.09 | 0.000 | 0.14 | 0.166 | 0.31 | 0.000 |
| | | 4 | 0.01 | 0.323 | 0.36 | 0.000 | 0.06 | 0.018 | 0.12 | 0.265 | 0.35 | 0.000 |
| | Alcohol consumption | 1 | 0.01 | 0.211 | 0.23 | 0.004 | 0.20 | 0.271 | 0.04 | 0.010 | 0.22 | 0.004 |
| | | 2 | 0.00 | 0.808 | 0.28 | 0.000 | -0.04 | 0.764 | 0.02 | 0.137 | 0.28 | 0.000 |
| | | 3 | 0.00 | 0.596 | 0.31 | 0.000 | 0.08 | 0.535 | 0.02 | 0.098 | 0.31 | 0.000 |
| | | 4 | 0.00 | 0.735 | 0.35 | 0.000 | 0.33 | 0.017 | 0.01 | 0.718 | 0.35 | 0.000 |
| | Exercise (units per week) | 1 | 0.00 | 0.798 | 0.22 | 0.004 | -0.08 | 0.340 | -0.01 | 0.731 | 0.22 | 0.004 |
| | | 2 | 0.00 | 0.613 | 0.28 | 0.000 | -0.10 | 0.133 | -0.02 | 0.544 | 0.28 | 0.000 |
| | | 3 | 0.00 | 0.278 | 0.31 | 0.000 | -0.11 | 0.059 | -0.03 | 0.294 | 0.31 | 0.000 |
| | | 4 | 0.00 | 0.866 | 0.35 | 0.000 | -0.10 | 0.114 | 0.01 | 0.832 | 0.35 | 0.000 |

*Table 4 continued*

| Outcome | Mediators | Sample time point | Indirect effect | | Total effect | | a | | b | | c | |
|---|---|---|---|---|---|---|---|---|---|---|---|---|
| | | | estimate | p | estimate | p | estimate | p | estimate | p | estimate | p |
| | | 1 | 0.01 | 0.085 | 0.23 | 0.003 | −0.08 | 0.110 | −0.15 | 0.005 | 0.22 | 0.004 |
| | | 2 | 0.00 | 0.392 | 0.28 | 0.000 | −0.02 | 0.332 | −0.19 | 0.023 | 0.28 | 0.000 |
| | | 3 | 0.01 | 0.371 | 0.31 | 0.000 | −0.17 | 0.000 | −0.05 | 0.350 | 0.31 | 0.000 |
| | Sleep (hours per night) | 4 | 0.01 | 0.303 | 0.36 | 0.000 | −0.09 | 0.002 | −0.10 | 0.278 | 0.35 | 0.000 |

a: pathway estimate between exogenous latent variable and mediator; b: pathway estimate between mediator and endogenous variable; c: pathway estimate between exogenous latent variable and endogenous variable.

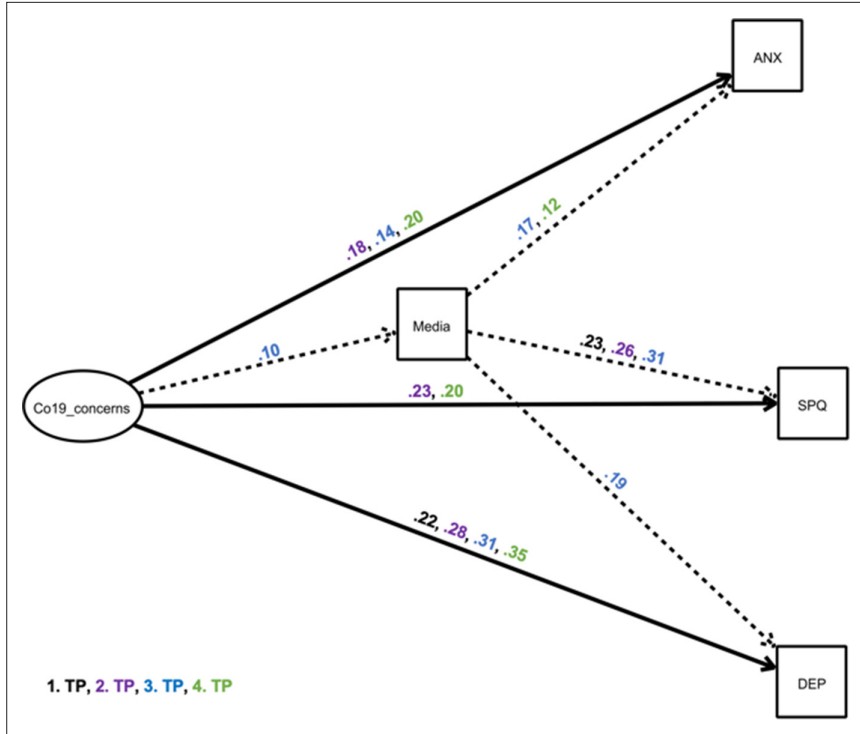

**Figure 4.** Structural equation model showing a mediator model of the impact of 'COVID-19 related life concerns' on depressive and anxiety-related symptoms and Schizotypal traits, via the mediator excessive media use. The exogenous latent variable ('predictor') had an elevating effect on the endogenous variable ('outcome') and the mediator. Black estimates: April/May 2020 survey, purple estimates: September/October 2020 survey, blue estimates: January/February 2021 survey, green estimates: May 2021 survey. Solid, bold lines indicate direct effects; dashed lines indicate indirect effects. Effects are only shown where p<0.05 (see **Table 4**). Other possible mediators were included in the model, but are not depicted for simplicity, as indirect effects were non-significant (**Table 4**). ANX: anxiety symptoms; DEP: depressive Symptoms, SPQ: total sum scores of Schizotypal Personality Questionnaire; TP: time point; Co19_concerns: COVID-19 related life concerns.

## 'Social adversity' Model

The results for this model are presented in **Table 5**, and for the mediator media in **Figure 5**.

The structural equation model revealed that the exogenous latent variable 'social adversity' was directly related to schizotypy, anxiety, and depression in all surveys (pathway c), and social adversity was positively associated with media (January/ February 2021) and drug (April/ May 2020, January/ February 2021, May 2021) consumption, and negatively associated with sleep (all surveys). Alcohol was positively correlated with social adversity at the April/ May 2020 survey: people who were more isolated drank more alcohol (pathway a). The alternative models revealed a worse model fit and are presented in the **Supplementary file 1n** for AIC/BIC in comparison.

Exploring the effect of 'social adversity' on schizotypy (**Figure 5**, **Table 5**), we found that in all four surveys, social adversity was associated with higher levels of schizotypal traits, with this association being most pronounced in the May 2021 survey (c=0.33). Significant total effects were shown in all four surveys. The mediation with excessive media consumption revealed a statistically significant indirect effect in the January/February 2021 survey: 11.5% of the effects of social adversity during the pandemic on schizotypy were mediated by excessive media consumption.

Investigating the effect of social adversity on anxiety (**Figure 5**, **Table 5**), structural equation models revealed that the exogenous latent variable was associated with heightened anxiety (c=0.25–0.29). All total effects were significant. The mediation with excessive media consumption revealed again a statistically significant indirect effect in the January/February 2021 survey: 3.6% of the effects of social adversity during the pandemic on depressive symptoms were mediated by excessive media use. Again, there was no other statistically significant indirect effect, revealing no additional mediators.

**Table 5.** Overview of all results of the structural equation model with 'social adversity' as an exogeneous latent variable for mental health endogenous variables with a mediation by harmful and healthy behaviours.

The results are separated by endogenous variable (schizotypal traits, depressive symptoms, and anxiety symptoms), mediators (excessive media use, drug consumption, alcohol consumption, units of exercise per week, and hours of sleep per night during weeks) and samples at the four time points, showing indirect, direct, and total effects, as well as individual pathways.

| Outcome | Mediators | Sample time point | Indirect effect estimate | p | Total effect estimate | p | a estimate | p | b estimate | p | c estimate | p |
|---|---|---|---|---|---|---|---|---|---|---|---|---|
| SPQ | Excessive Media use | 1 | 0.01 | 0.156 | 0.13 | 0.027 | 0.05 | 0.062 | 0.21 | 0.022 | 0.12 | 0.043 |
| | | 2 | 0.01 | 0.219 | 0.25 | 0.000 | 0.04 | 0.122 | 0.25 | 0.015 | 0.24 | 0.001 |
| | | 3 | 0.03 | 0.017 | 0.26 | 0.000 | 0.11 | 0.000 | 0.25 | 0.004 | 0.23 | 0.000 |
| | | 4 | 0.00 | 0.989 | 0.33 | 0.000 | 0.05 | 0.056 | 0.00 | 0.987 | 0.33 | 0.000 |
| | Drug consumption | 1 | 0.01 | 0.383 | 0.13 | 0.029 | 0.07 | 0.001 | 0.13 | 0.352 | 0.12 | 0.043 |
| | | 2 | 0.01 | 0.313 | 0.25 | 0.000 | 0.05 | 0.077 | 0.18 | 0.152 | 0.24 | 0.001 |
| | | 3 | 0.01 | 0.700 | 0.24 | 0.000 | 0.10 | 0.000 | 0.05 | 0.699 | 0.23 | 0.000 |
| | | 4 | 0.01 | 0.455 | 0.33 | 0.000 | 0.06 | 0.017 | 0.11 | 0.403 | 0.33 | 0.000 |
| | Alcohol consumption | 1 | 0.00 | 0.903 | 0.12 | 0.038 | 0.50 | 0.000 | 0.13 | 0.352 | 0.12 | .043 |
| | | 2 | 0.00 | 0.805 | 0.24 | 0.001 | -0.08 | 0.570 | 0.01 | 0.577 | 0.24 | .001 |
| | | 3 | 0.00 | 0.477 | 0.23 | 0.000 | 0.16 | 0.209 | -0.02 | 0.288 | 0.23 | .000 |
| | | 4 | 0.00 | 0.810 | 0.33 | 0.000 | 0.05 | 0.693 | -0.02 | 0.385 | 0.33 | .000 |
| | Exercise (units per week) | 1 | 0.00 | 0.785 | 0.12 | 0.047 | 0.03 | 0.715 | -0.04 | 0.202 | 0.12 | .043 |
| | | 2 | 0.00 | 0.971 | 0.24 | 0.001 | -0.06 | 0.454 | 0.00 | 0.951 | 0.24 | .001 |
| | | 3 | 0.00 | 0.968 | 0.23 | 0.000 | -0.03 | 0.587 | 0.00 | 0.934 | 0.23 | .000 |
| | | 4 | 0.00 | 0.606 | 0.33 | 0.000 | -0.05 | 0.434 | 0.05 | 0.279 | 0.33 | .000 |
| | Sleep (hours per night) | 1 | 0.01 | 0.257 | 0.13 | 0.020 | -0.16 | 0.000 | -0.09 | 0.217 | 0.12 | .043 |
| | | 2 | 0.02 | 0.131 | 0.26 | 0.000 | -0.11 | 0.000 | -0.23 | 0.115 | 0.24 | .001 |
| | | 3 | 0.01 | 0.223 | 0.25 | 0.000 | -0.13 | 0.001 | -0.10 | 0.196 | 0.23 | .000 |
| | | 4 | -0.01 | 0.736 | 0.32 | 0.000 | -0.10 | 0.000 | 0.05 | 0.717 | 0.33 | .000 |
| Anxiety | Excessive Media use | 1 | 0.00 | 0.340 | 0.26 | 0.000 | 0.05 | 0.062 | 0.06 | 0.226 | 0.25 | .000 |
| | | 2 | 0.00 | 0.286 | 0.27 | 0.000 | 0.04 | 0.122 | 0.09 | 0.116 | 0.27 | .000 |
| | | 3 | 0.01 | 0.021 | 0.28 | 0.000 | 0.11 | 0.000 | 0.13 | 0.012 | 0.26 | .000 |
| | | 4 | 0.00 | 0.192 | 0.29 | 0.000 | 0.05 | 0.056 | 0.09 | 0.058 | 0.29 | .000 |
| | Drug consumption | 1 | 0.00 | 0.918 | 0.25 | 0.000 | 0.07 | 0.001 | -0.01 | 0.915 | 0.25 | .000 |
| | | 2 | 0.01 | 0.404 | 0.27 | 0.000 | 0.05 | 0.077 | 0.09 | 0.247 | 0.27 | .000 |
| | | 3 | 0.01 | 0.307 | 0.27 | 0.000 | 0.10 | 0.000 | 0.08 | 0.307 | 0.26 | .000 |
| | | 4 | 0.00 | 0.795 | 0.29 | 0.000 | 0.06 | 0.017 | -0.02 | 0.777 | 0.29 | .000 |

*Table 5 continued on next page*

*Table 5 continued*

| Outcome | Mediators | Sample time point | Indirect effect | | Total effect | | a | | b | | c | |
|---|---|---|---|---|---|---|---|---|---|---|---|---|
| | | | estimate | p | estimate | p | estimate | p | estimate | p | estimate | p |
| | Alcohol consumption | 1 | 0.00 | 0.513 | 0.25 | 0.000 | 0.50 | 0.000 | −0.01 | 0.487 | 0.25 | .000 |
| | | 2 | 0.00 | 0.936 | 0.27 | 0.000 | −0.08 | 0.570 | 0.00 | 0.861 | 0.27 | .000 |
| | | 3 | 0.00 | 0.526 | 0.26 | 0.000 | 0.16 | 0.209 | −0.01 | 0.324 | 0.26 | .000 |
| | | 4 | 0.00 | 0.842 | 0.29 | 0.000 | 0.05 | 0.693 | 0.01 | 0.511 | 0.29 | .000 |
| | Exercise (units per week) | 1 | 0.00 | 0.982 | 0.25 | 0.000 | 0.03 | 0.715 | 0.00 | 0.951 | 0.25 | .000 |
| | | 2 | 0.00 | 0.872 | 0.27 | 0.000 | −0.06 | 0.454 | −0.01 | 0.784 | 0.27 | .000 |
| | | 3 | 0.00 | 0.988 | 0.26 | 0.000 | −0.03 | 0.587 | 0.00 | 0.976 | 0.26 | .000 |
| | | 4 | 0.00 | 0.485 | 0.28 | 0.000 | −0.05 | 0.434 | 0.05 | 0.011 | 0.29 | .000 |
| | Sleep (hours per night) | 1 | 0.00 | 0.539 | 0.26 | 0.000 | −0.16 | 0.000 | −0.02 | 0.524 | 0.25 | .000 |
| | | 2 | 0.00 | 0.847 | 0.27 | 0.000 | −0.11 | 0.000 | −0.01 | 0.843 | 0.27 | .000 |
| | | 3 | 0.00 | 0.627 | 0.26 | 0.000 | −0.13 | 0.001 | 0.02 | 0.620 | 0.26 | .000 |
| | | 4 | 0.00 | 0.623 | 0.28 | 0.000 | −0.10 | 0.000 | 0.03 | 0.597 | 0.29 | .000 |
| Depression | Excessive Media use | 1 | 0.00 | 0.489 | 0.48 | 0.000 | 0.05 | 0.062 | 0.04 | 0.430 | 0.47 | .000 |
| | | 2 | 0.00 | 0.464 | 0.47 | 0.000 | 0.04 | 0.122 | 0.06 | 0.353 | 0.46 | .000 |
| | | 3 | 0.01 | 0.024 | 0.51 | 0.000 | 0.11 | 0.000 | 0.13 | 0.019 | 0.49 | .000 |
| | | 4 | 0.00 | 0.393 | 0.56 | 0.000 | 0.05 | 0.056 | 0.06 | 0.333 | 0.55 | .000 |
| | Drug consumption | 1 | 0.00 | 0.730 | 0.47 | 0.000 | 0.07 | 0.001 | −0.03 | 0.708 | 0.47 | .000 |
| | | 2 | 0.01 | 0.383 | 0.47 | 0.000 | 0.05 | 0.077 | −0.10 | 0.279 | 0.46 | .000 |
| | | 3 | 0.00 | 0.662 | 0.50 | 0.000 | 0.10 | 0.000 | 0.04 | 0.664 | 0.49 | .000 |
| | | 4 | 0.01 | 0.362 | 0.56 | 0.000 | 0.06 | 0.017 | 0.09 | 0.287 | 0.55 | .000 |
| | Alcohol consumption | 1 | 0.00 | 0.667 | 0.48 | 0.000 | 0.50 | 0.000 | 0.01 | 0.663 | 0.47 | .000 |
| | | 2 | 0.00 | 0.670 | 0.46 | 0.000 | −0.08 | 0.570 | 0.02 | 0.102 | 0.46 | .000 |
| | | 3 | 0.00 | 0.373 | 0.50 | 0.000 | 0.16 | 0.209 | 0.02 | 0.202 | 0.49 | .000 |
| | | 4 | 0.00 | 0.747 | 0.55 | 0.000 | 0.05 | 0.693 | −0.02 | 0.149 | 0.55 | .000 |
| | Exercise (units per week) | 1 | 0.00 | 0.761 | 0.47 | 0.000 | 0.03 | 0.715 | −0.03 | 0.113 | 0.47 | .000 |
| | | 2 | 0.00 | 0.571 | 0.47 | 0.000 | −0.06 | 0.454 | −0.03 | 0.212 | 0.46 | .000 |
| | | 3 | 0.00 | 0.621 | 0.50 | 0.000 | −0.03 | 0.587 | −0.04 | 0.079 | 0.49 | .000 |
| | | 4 | 0.00 | 0.876 | 0.55 | 0.000 | −0.05 | 0.434 | −0.01 | 0.795 | 0.55 | .000 |

*Table 5 continued*

| Outcome | Mediators | Sample time point | Indirect effect | | Total effect | | a | | b | | c | |
|---|---|---|---|---|---|---|---|---|---|---|---|---|
| | | | estimate | p | estimate | p | estimate | p | estimate | p | estimate | p |
| | | 1 | 0.01 | 0.384 | 0.48 | 0.000 | −0.16 | 0.000 | −0.03 | 0.371 | 0.47 | .000 |
| | | 2 | −0.01 | 0.463 | 0.46 | 0.000 | −0.11 | 0.000 | 0.06 | 0.431 | 0.46 | .000 |
| | Sleep (hours per Night) | 3 | 0.00 | 0.814 | 0.50 | 0.000 | −0.13 | 0.001 | −0.01 | 0.807 | 0.49 | .000 |
| | | 4 | 0.00 | 0.868 | 0.55 | 0.000 | −0.10 | 0.000 | 0.03 | 0.597 | 0.55 | .000 |

a: pathway estimate between exogenous latent variable and mediator; b: pathway estimate between mediator and endogenous variable; c: pathway estimate between exogeneous latent variable and endogenous variable.

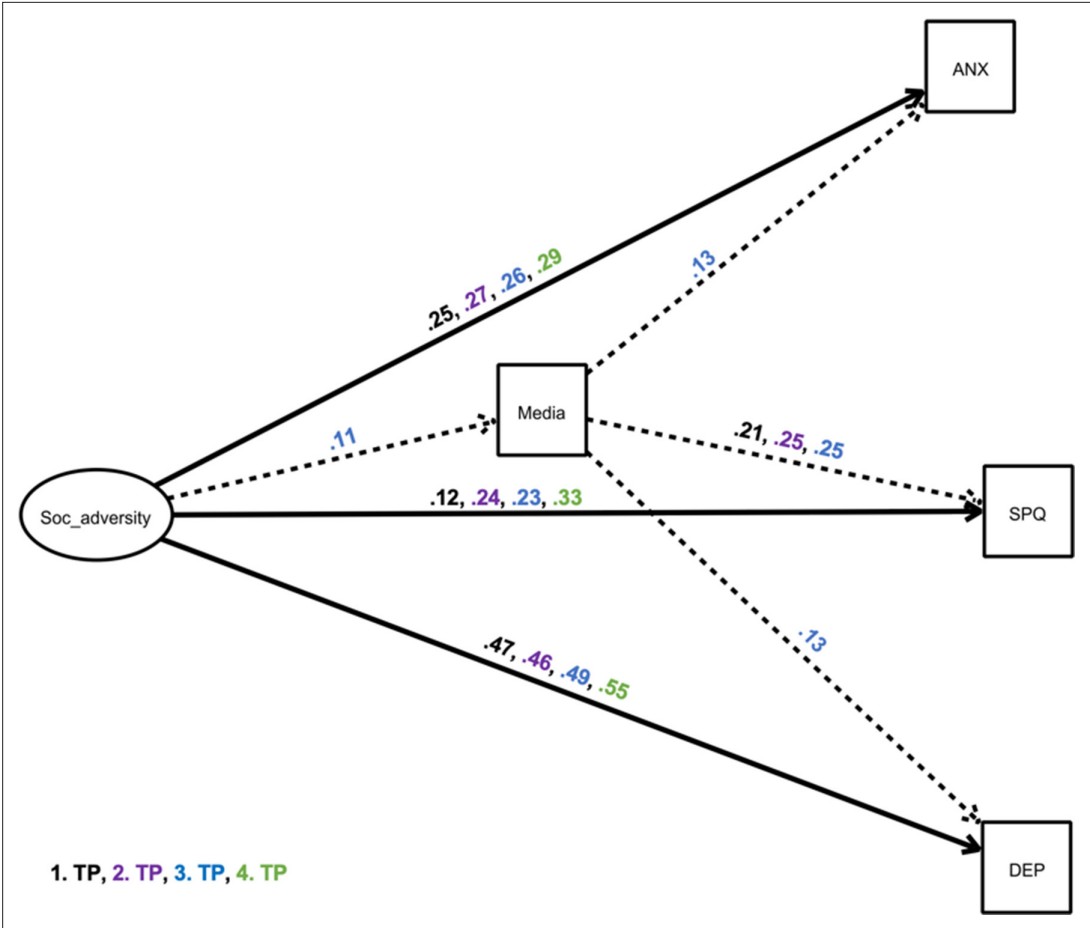

**Figure 5.** Structural equation model showing a mediator model of the impact of 'social adversity' on depressive and anxiety-related symptoms and schizotypal traits and via the mediator excessive media use. The exogeneous latent variable ('predictor') had an elevating effect on the endogenous variable ('outcome') and the mediator. We only present significant pathways (p<0.05). Black estimates: April/May 2020 survey, purple estimates: September/October 2020 survey, blue estimates: January/February 2021, green estimates: May 2021 survey. Solid, bold lines indicate direct effects; dashed lines indicate indirect effects. Other possible mediators were included in the model, but are not depicted for simplicity, as indirect effects were non-significant (*Table 5*). ANX: anxiety symptoms; DEP: depressive Symptoms, SPQ: total sum scores of Schizotypal Personality Questionnaire; TP: time point; Soc_adversity: social adversity.

Furthermore, SEM revealed that the exogeneous latent variable 'social adversity' had a positive association with depressive symptoms in all four surveys (*Figure 5*, *Table 5*). All total effects were significant. The mediation with excessive media consumption revealed again a statistically significant indirect effect in the January/February 2021 survey, with 2% of effects explained.

## Exploratory model: mediating effects of anxiety and depression on schizo-typal traits

In an exploratory model, we investigated the mediating effect of anxiety and depression on the association between COVID-19 related life concerns and schizotypy, and social adversity and schizotypy. The model fit is presented in *Supplementary file 1o*. We found that the association between COVID-19 related life concerns and schizotypy was fully mediated by anxiety scores only in the January/February 2021 and May 2021 surveys. Anxiety also mediated the effects of social adversity and schizotypy across all four surveys (*Table 6*). Depression did not act as a mediator between COVID-19 related life concerns and schizotypy, but it did mediate effects of social adversity predicting schizotypy in January/February 2021.

**Table 6.** Exploratory analysis: Anxiety and Depression as Mediators and Schizotypy as Outcome.

| Predictor | Mediators | Sample time point | Indirect effect estimate | p | Total effects estimate | p | a estimate | p | b estimate | p | c estimate | p |
|---|---|---|---|---|---|---|---|---|---|---|---|---|
| Co19 related life concerns | Depression | 1 | 0.00 | 0.999 | -0.03 | 0.637 | 0.24 | 0.001 | 0.00 | 0.998 | -0.03 | .738 |
| | | 2 | 0.02 | 0.458 | 0.08 | 0.127 | 0.29 | 0.000 | 0.07 | 0.457 | 0.06 | .299 |
| | | 3 | 0.07 | 0.070 | -0.01 | 0.895 | 0.34 | 0.000 | 0.21 | 0.013 | -0.08 | .157 |
| | | 4 | 0.07 | 0.051 | 0.03 | 0.623 | 0.37 | 0.000 | 0.18 | 0.035 | -0.04 | .558 |
| | Anxiety | 1 | 0.05 | 0.092 | 0.03 | 0.734 | 0.08 | 0.101 | 0.68 | 0.000 | -0.03 | .738 |
| | | 2 | 0.09 | 0.209 | 0.35 | 0.002 | 0.19 | 0.000 | 0.80 | 0.000 | 0.06 | .299 |
| | | 3 | 0.16 | 0.000 | 0.08 | 0.164 | 0.17 | 0.000 | 0.93 | 0.000 | -0.08 | .157 |
| | | 4 | 0.17 | 0.000 | 0.13 | 0.041 | 0.20 | 0.000 | 0.87 | 0.000 | -0.04 | .558 |
| Social adversity | Depression | 1 | 0.05 | 0.531 | -0.03 | 0.571 | 0.48 | 0.000 | 0.11 | 0.527 | -0.08 | .442 |
| | | 2 | 0.08 | 0.249 | 0.05 | 0.371 | 0.46 | 0.000 | 0.17 | 0.567 | -0.03 | .787 |
| | | 3 | 0.17 | 0.004 | 0.01 | 0.789 | 0.51 | 0.000 | 0.34 | 0.002 | -0.16 | .028 |
| | | 4 | 0.10 | 0.145 | 0.09 | 0.142 | 0.56 | 0.000 | 0.17 | 0.146 | -0.01 | .896 |
| | Anxiety | 1 | 0.18 | 0.000 | 0.10 | 0.342 | 0.26 | 0.000 | 0.71 | 0.000 | -0.08 | .442 |
| | | 2 | 0.23 | 0.000 | 0.21 | 0.049 | 0.28 | 0.000 | 0.83 | 0.000 | -0.03 | .787 |
| | | 3 | 0.26 | 0.000 | 0.10 | 0.216 | 0.28 | 0.000 | 0.94 | 0.000 | -0.16 | .028 |
| | | 4 | 0.24 | 0.000 | 0.23 | 0.012 | 0.29 | 0.000 | 0.85 | 0.000 | -0.01 | .896 |

a: pathway estimate between exogeneous latent variable and mediator; b: pathway estimate between mediator and endogenous variable; c: pathway estimate between exogeneous latent variable and endogenous variable.

## Discussion

In this paper, we investigated whether 'social adversity' and 'COVID-19 related life concerns' predicted schizotypal traits, anxiety, and depression in snowball-ascertained samples of British and German participants at four time points throughout the pandemic. We furthermore explored whether these associations were mediated by healthy and unhealthy behaviours using a structural equation model-ling approach.

### COVID-19 related life concerns' model

Our results showed that COVID-19 related life concerns were associated with SPQ scores in September/October 2020 and May 2021, anxiety symptoms in all surveys except the first lockdown in March 2020, and depressive symptoms in all four surveys. Interestingly, COVID-19 related concerns were associated with schizotypal traits in those samples collected when the restrictions on daily life and the restrictive measures were less severe (September/October 2020 *Daimer et al., 2021* and May 2021 samples). This may be attributed to characteristics of a schizotypal personality. While some people may be less stressed due to lower restrictive measures and may quickly return to normality, others with more schizotypal traits may still be suspicious of the situation and unable to trust it (*Preti et al., 2020*), whereas they were more likely to feel secure when restrictions were in place, which is why effects were not seen during lockdowns.

COVID-19 related concerns, moreover, predicted an increase in media consumption, in the samples measured at the January/February 2021 survey, confirming that individuals who experienced high stress related to COVID-19 may use excessive media consumption as a potential coping strategy (*Bendau et al., 2021*). Media consumption was also associated with increased schizotypy and depres-sive and anxiety symptoms in people experiencing high levels of COVID-19 related concerns in the sample measured in January/February 2021, when case numbers were highest in both countries. Here we found that media consumption significantly mediated the effect of COVID-19 related life concerns on schizotypy and depressive and anxiety symptoms, potentially by increasing levels of fear and misin-formation (*Tasnim et al., 2020*) and thus increasing these symptoms (*Ni et al., 2020*; *Satici et al., 2020*).

COVID-19 related life concerns also predicted drug consumption in the samples from September/ October 2020, January/February 2021 and May 2021, indicating the role of drugs as a coping strategy of burdened people (*Czeisler et al., 2020*). Furthermore, a positive correlation was found between COVID-19 related life concerns and alcohol consumption only in the May 2021 survey. This could be due to the calming effect of alcohol on the nervous system, hence it could be expected that consumption might increase in response to stressors (*Kypri and McCambridge, 2018*; *Eckardt et al., 1998*). However, we did not find this effect in the three other surveys. Interestingly, general alcohol consumption declined in most of the European countries, with the exception of the UK (*Kilian et al., 2021*) during the pandemic.

Our first model also revealed that higher levels of COVID-19 related life concerns was associated with fewer hours of sleep at night in the January/February 2021 and May 2021 surveys. Poor sleeping or poor sleeping patterns (e.g.<6 hr of sleep) are associated with worse mental health (*Becker et al., 2018*; *João et al., 2018*). Franceschini and colleagues (*Franceschini et al., 2020*) showed that 55% of 6519 adults reported poor sleep quality during the first lockdown, which was associated with higher levels of stress during the pandemic. Consistent with these findings, we found that more hours of sleep (>6 hr) were associated with lower levels of depression in the April/May 2020 and September/ October 2020 surveys, lower anxiety levels in the April/May 2020 survey and lower SPQ scores in September/October 2020. Sleep may therefore play an important role as an emotional stabiliser (*Goldstein and Walker, 2014*).

We did not find a negative association between COVID related life concerns and physical activity; however, we found a positive association between physical activity and mental health scores; indicating that more physical activity is associated with higher anxiety in the September/October 2020 survey and May 2021 survey. This is surprising and contradicts a large body of literature showing the positive effect of exercise on levels of anxiety, depression, and stress (for review see *Mikkelsen et al., 2017*). However, people who play sports frequently may have suffered more from the pandemic containment measures, which included severe restrictions on the execution of sports. Here, especially team sports are affected, which combines positive social interaction with sports. In order to compensate, affected

individuals may over-compensate with individual sports which lack the social component. The associations found in our surveys occurred when general restrictions were lower, therefore, more exercise potentially means an increased risk of exposure to potentially infected individuals, for example, in gyms. A study by Mehrsafar et al. showed that among professional athletes, isolation from their athletic team, reduced activity and training, lack of formal coaching, and lack of social support from fans and media triggered emotional distress (*Mehrsafar et al., 2020*). A third possibility could be that individuals started exercising more frequently and regularly during the pandemic as a result of loneliness, boredom or the knowledge of positive effects of sports on anxiety and depression; however, sport alone cannot completely protect against mental health problems (*Pensgaard et al., 2021*).

A potential reason for why we did not see a general positive effect might be the sensitivity of the measure (number of times of at least 30 min increased heart rate activity per week). Also, individuals may have changed their exercise habits from, for example, team sport to individual exercise, which may dampen the positive impact.

## Social adversity model

In the second model, we explored and investigated the association between social adversity or the lack of supportive relationships and mental health, and how those associations were mediated by healthy and unhealthy behaviour. We found that in all four surveys, social adversity predicted the expression of schizotypal traits. A study by *Le et al., 2019* found that in individuals with trait schizotypy, psychosis-like symptoms were exacerbated only in those who experienced increased loneliness. Individuals high in schizotypal traits tend to have fewer social skills. Therefore, an increase in loneliness during the pandemic might heighten schitzotypal traits, potentially pushing individuals with already high scores of schizotypy into full-blown psychosis (*Brown et al., 2020*; *Chau et al., 2019*; *Esposito et al., 2021*). Our findings, in the light of the literature, show the tremendous need for enabling safe social contacts during situations which require social distancing. Furthermore, we found that social stress predicted both depressive and anxiety symptoms in all four samples. This is consistent with the social exclusion theory, which identifies social exclusion as the primary source of anxiety (*Leary, 1990*). Other studies during the COVID-19 pandemic have also shown that loneliness was the main risk factor for depression and anxiety (*Okruszek et al., 2020*; *Palgi et al., 2020*). *McQuaid et al., 2021* found that loneliness related to anxiety and depression increased in a dose-like fashion during the first half of the COVID-19 pandemic.

In addition, social adversity was associated with increased excessive media use in the January/February 2021 sample and increased drug use in sample at the surveys from April/May 2020, January/February 2021 and May 2021. Both have been discussed as maladaptive coping mechanisms to reduce loneliness (*Rokach, 2005*). Increased media consumption during the pandemic has been described as a compensation for the lack of social contacts (*Cauberghe et al., 2021*). However, excessive use of social media increases the risk of misinformation (*Ahmed Siddiqui et al., 2020*; *Tasnim et al., 2020*), and as pointed out above is linked to lower mental health, especially to higher levels of anxiety and depression (*Gao et al., 2020*; *Zhao and Zhou, 2020*). Furthermore, higher social adversity also predicted higher alcohol consumption in the April/May 2020 survey. This showed that people who felt more lonely or isolated at the beginning of the pandemic also drank more alcohol, as previously found (*Horigian et al., 2021*). Similar to the first model, social adversity was found to reduce sleep. Research on sleep, social adversity and mental health is inconsistent, with some studies reporting no influence of social adversity or loneliness on sleep (*Hawkley et al., 2010*), while others show a clear negative impact of social adversity and loneliness on sleep (*Bartoszek et al., 2020*; *Groarke et al., 2020*). Unhealthy behavioural patterns might act in a negative self-reinforcing way, as it has been reported that lack of sleep leads people to participate less in social interactions and to withdraw (*Ben Simon and Walker, 2018*), increasing the risk for reduced mental health, which might again negatively impact the quality of sleep. However, our cross-sectional data does not show evidence of a direct link between reduced sleep, schizotypy, depressive or anxiety symptoms. We also found no association between social adversity and physical activity. This contradicts the results of other studies that report that loneliness decreases the likelihood of physical activity (*Creese et al., 2021*; *Hawkley et al., 2009*). However, we again found that more exercise was associated with higher levels of anxiety in the May 2021 survey (see possible explanation in section above).

Excessive media consumption partially mediated the effects of social adversity on SPQ traits. Thus, individuals who suffer more from loneliness or lack of social support have higher schizotypy in part because of their propensity for high levels of media consumption. However, we did not find mediation of the social adversity and schizotypy association via drugs, alcohol consumption, exercise, or sleep.

In exploratory analyses, we investigated the mediating effect of anxiety and depressive symptoms on the association between social adversity and schizotypal traits and COVID-19 related concerns and schizotypal traits. Depressive symptoms did not mediate the effects of COVID-19 concerns on schizotypy, but they did mediate the effects of social adversity on schizotypy, but only during the January/February 2021 survey. We found that the association between social adversity as well as COVID-19 related concerns and schizotypy was in both cases fully mediated by anxiety (all surveys for the former effects, only January/February 2021 and May 2021 for the latter effects). This means that social adversity and COVID-19 related concerns increased anxiety in our samples, which in turn impacted the level of schizotypal traits. Findings are consistent with other work reporting a positive association between schizotypy and social anxiety (*Brown et al., 2008*; *Lewandowski et al., 2006*).

There are several limitations to our study. First, in this analysis, we investigate four different samples collected at different time points within 1 year from the start of the COVID-19 pandemic. Although the samples are highly comparable, and we adjust for observed differences between them in our models, we cannot adjust for unmeasured confounding. Thus, we cannot definitively say that the altered associations observed throughout the pandemic are related to genuine changes in these relationships and thus reflect the changing impact of the pandemic.

Second, the sample might be biased due to the recruitment strategy. Recruitment was performed through print and social media; however, the questionnaire was only available online, so people without internet access or less ability in using the internet were either excluded from participation or had to rely on people guiding them through the survey. This especially applies to older individuals, who have less access to the internet than young people (*Prescott, 2021*; *Quittschalle et al., 2020*). However, we were able to recruit individuals from ages 18–93, with 16.73% aged above 60, which shows a good general representation of age. Since excessive media consumption was investigated in this study, the recruitment strategy, especially via the Internet, may have led to a bias, reaching a disproportionately large number of people who use the media excessively. However, in the first investigation (*Knolle et al., 2021*) using the same recruitment strategy, we found that media consumption increased during the COVID-19 pandemic compared to prior to the pandemic. Also, a sampling bias may have occurred over-representing individuals attracted to the topic of mental health (*Andrade, 2020*), which could on one hand heighten the strength of the observed associations compared with a representative general population sample and on the other explain the overrepresentation of individuals with higher educational backgrounds. Biases like these are difficult to overcome, especially in psychological research. These limitations may affect the generalisability and representativeness of the study.

Third, an adequate fit is assumed for RMSEA of less than 0.06 and a CFI of greater than 0.90. Our models fulfil these criteria for the RMSEA, but are slightly lower for the CFI, ranging between 0.78 and 0.86. Our models are highly complex, with each model including five mediators and three outcomes. When reducing the complexity of the models, fit based on CFI meets threshold (≥0.9); however, for the sake of parsimony we present the full models.

Important implications can be drawn from the results for possible future pandemics and national or global health emergencies in order to mitigate a drastic deterioration of the mental health of the general population. To prevent severe isolation and loneliness, social contact should be allowed in secure and low-risk settings, such as meeting outside in pairs, wearing masks and, if necessary, keeping some distance. In general, people who live alone are particularly affected by loneliness, which is a major risk factor for experiencing many negative mental states, including psychosis-like experiences (*Butter et al., 2017*; *Le et al., 2019*). A recent study has shown that anxiety can be strongly reduced by half by the mere presence of another person (*Qi et al., 2020*). Furthermore, as excessive media consumption exacerbates the negative effects of COVID-19 related stress and social adversity, it is important to educate individuals on healthy and responsible media consumption. It is necessary to implement these educational measures in school curriculums and professional training settings. During the pandemic we saw an increase in mental health crises, and symptoms of depression and anxiety. Healthcare providers or city councils could offer free online sessions to foster resilience and

mental well-being, such as mindfulness, which has been shown to improve general mental health in adults (*Antesberger et al., 2021*). Critically, special attention should be paid to people with pre-existing mental health conditions or at high risk of developing disorders such as psychosis, as such individuals are more likely to experience exacerbating symptoms during challenging circumstances such as the COVID-19 pandemic (*Fekih-Romdhane et al., 2021*; *Iob et al., 2020*; *Preti et al., 2020*). Furthermore, expansion of online psychiatric interventions could be beneficial, as these have shown promise in ameliorating symptoms (*Cheli et al., 2020*; *DeLuca et al., 2020*; *Schleider et al., 2022*).

In conclusion, we found that social adversity and COVID-19 related concerns predicted higher schizotypal traits, and symptoms of depression and anxiety. We furthermore found that excessive media consumption (more than 4 hr a day) partially mediated these relationships. We identified the ameliorating potential of regular sleep on mental health, but especially on schizotypy. Overall, our study shows that during the handling of extreme situations such as a global pandemic which require lockdowns and social distancing, sustained meaningful relationships, and a healthy lifestyle are essential to maintain mental health. Furthermore, our findings highlight the risk of pandemic stressors heightening people's levels of schizotypy, especially through excessive media consumption. Finally, our findings show that symptoms of common mental disorders (depression and anxiety) mediated the effects of social adversity and COVID-19 related concerns on schizotypy. All together our findings underscore the need for protective measures to be in place, such as social support networks, psycho-education, media education, and treatment for common mental disorders, to support those most vulnerable to increased schizotypy.

## Acknowledgements

We thank the participants of the study who contributed their time and further circulated the survey.

## Additional information

### Funding

| Funder | Grant reference number | Author |
|--------|------------------------|--------|
| Horizon 2020 Framework Programme | 754462 | Franziska Knolle |
| Cundill Centre for Child and Youth Depression | | Sharon AS Neufeld |
| Wellcome Trust Institutional Strategic Support Fund | 204845/Z/16/Z | Sharon AS Neufeld |

The funders had no role in study design, data collection and interpretation, or the decision to submit the work for publication.

### Author contributions

Sarah Daimer, Conceptualization, Formal analysis, Investigation, Methodology, Project administration, Supervision, Writing - original draft, Writing – review and editing; Lorenz L Mihatsch, Sharon AS Neufeld, Methodology, Writing – review and editing; Graham K Murray, Conceptualization, Writing – review and editing; Franziska Knolle, Conceptualization, Formal analysis, Funding acquisition, Methodology, Project administration, Writing - original draft

### Author ORCIDs

Sarah Daimer ⓘ http://orcid.org/0000-0002-9291-0823
Lorenz L Mihatsch ⓘ http://orcid.org/0000-0003-3835-7964
Sharon AS Neufeld ⓘ http://orcid.org/0000-0001-5470-3770
Franziska Knolle ⓘ http://orcid.org/0000-0002-9542-613X

### Ethics

Ethical approval was obtained from the Ethical Commission Board of the Technical University Munich (250/20 S). The ethical approval was provided for data collection in and outside Germany; all data

security measures are met and is stored exclusively on a server of the Technical University of Munich, in Germany. All respondents included in the analyses provided informed consent.

## Decision letter and Author response
Decision letter https://doi.org/10.7554/eLife.75893.sa1
Author response https://doi.org/10.7554/eLife.75893.sa2

---

## Additional files

### Supplementary files
• Supplementary file 1. Investigating the relationship of COVID-19 related stress and media consumption with schizotypy, depression, and anxiety in cross-sectional surveys repeated throughtout the pandemic in Germany and the UK - Supplementary file. (a) Overview of the model fit indices for predictor models separated by time point. (b) Results of Generalised Estimate Equation models for the three outcome variables with survey time point as predictor. (c) Pearson's correlation coefficient between SPQ sumscore, anxiety score (SCL-27) and depression score (SCL-27) at all four survey time points. (d) Overview of the model fit indices separated by exogenous latent variable and survey excluding 27 subjects, who completed all four time points. (e) Complete outcome of structural equation with COVID-19 related life concerns from first survey timepoint. (f) Complete outcome of structural equation with COVID-19 related life concerns from second survey time point. (g) Complete outcome of structural equation with COVID-19 related life concerns from third survey time point. (h) Complete outcome of structural equation with COVID-19 related life concerns from fourth survey time point. (i) Complete outcome of structural equation with Social adversity as predictor from the first survey time point. (j) Complete outcome of structural equation with Social adversity as predictor from second survey time point. (k) Complete outcome of structural equation with Social adversity as predictor from third survey time point. (l) Complete outcome of structural equation with Social adversity as predictor from fourth survey time point. (m) Model fit for model with reduced complexity, one predictor, one outcome, five mediators, example for time point 1. (n) Model comparison for original and alternative model. (o) Overview of the model fit indices separated by exogenous latent variable and time point.

• Transparent reporting form

### Data availability
Data is available for download: https://zenodo.org/record/5793973#.YcCuhi8w2Ak.

---

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
