## [Editor Report]

Using online surveys from Germany and the U.K. participants, this study examined the association between COVID-19 and mental health mediated by participants' behaviors. The results indicate that the pandemic was substantially associated with depressive and anxiety symptoms. Furthermore, unhealthy behaviors such as excessive media consumption further exacerbated the negative impact of social isolation due to COVID.

---

## [Decision Letter]

**Decision letter after peer review:**

Thank you for submitting your article "Investigating the relationship of COVID-19 related stress and media consumption with schizotypy, depression and anxiety in a cross-sectional study" for consideration by *eLife*. Your article has been reviewed by 2 peer reviewers, and the evaluation has been overseen by a Reviewing Editor and Ma-Li Wong as the Senior Editor. The following individuals involved in the review of your submission have agreed to reveal their identity: Sina Azadnajafabad (Reviewer #1); Rajini Nagrani (Reviewer #2).

Essential revisions:

1) Methodologically, I would suggest conducting a sensitivity analysis. Pool the data from all four time points together. You can use generalized equation modeling (GEE) to address repeated measures. To examine the association between the time of the survey and the outcome variables, the date of the survey can be a categorical variable (with April/May 2020 being the reference category) in the model. It is a more elegant way of presenting the results and enables you to statistically compare the outcomes across different periods during the pandemic (using marginal effect).

2) Correlation value (r) of less than 0.7 – even if the p-value is small (< 0.05) – does not mean a strong correlation between two variables. Please correct the reference on Page 16-17 of the manuscript.

3) Please address some of the discrepancies noted by reviewer #3 [i.e., reference to the Supplementary files, timing of the survey (months vs. season)].

4) Expand the limitation section of the paper to discuss: (a) lack of generalizability; and (b) use of social media for recruitment.

(5) Format the Supplementary file to be user-friendly (i.e., put them in word or excel tables).

*Reviewer #1 (Recommendations for the authors):*

I read the manuscript with interest as the topic was exciting. The manuscript is comprehensive in all sections. However, some comments need to be considered prior to any decision. My comments are as follows:

1. Title: Reflecting the sample size in two countries in the title of the manuscript could better show its scope.

2. Abstract: The methods part of the abstract is too short and needs to be expanded for a better understanding of the results part in the abstract. For example, the criteria or definition of the included conditions of schizotypy, depression, and anxiety.

3. Introduction: This section is very well prepared and introduces the issue in an adequate manner.

4. Methods: Although the authors referred to a publication regarding the questionnaire, it is highly suggested that authors briefly introduce the process of preparation and validation of this questionnaire for two countries of Germany and the UK, in the first section of the methods.

5. Methods: The used term "robust ANOVAS" in the statistical analysis section is somehow vague and needs further clarification.

6. Results, figures, and tables: Very well prepared and summarized.

7. Discussion: One paragraph is needed on the public health implications of the conducted investigation and expansion on the practical measures for policy-makers and health authorities.

*Reviewer #2 (Recommendations for the authors):*

1. I am not certain if it's okay to have ethics approval from only one participating country (Germany) for participants enrolled in the UK and Germany.

2. Though the authors mention their strategy of recruitment for follow-up surveys, it is unclear whether the same recruitment strategy was followed for first survey too.

3. It is not clear if the 26 participants which participated in all 4 surveys were included or excluded from the analysis. If included, authors are suggested to perform sensitivity analysis after excluding these participants.

4. Supplementary tables are not correctly cited in the text. For e.g.: there is no table in supplementary material titled as supplementary table 4 but is cited in the main text. Also it is confusing to interpret what authors mean by Table X in supplementary material. Authors are suggested not to use titles such as "Table 4" and "Table 2" in supplementary material instead prefix them with word "supplementary". At two instances authors have vaguely mentioned supplementary materials in section 3.3, authors are suggested to cite the supplementary table no. they are referring to. Finally supplementary tables should appear in the order citied in the text (for example suppl table 2 appears before suppl table 1 in text).

5. Authors use some words interchangeably, for example 1: Though authors clarify the usage of gender (and not biological sex), this has been interchangeably mentioned in section 2.4, pg 13. Example 2: Authors use the words may 2021 and spring 2021 interchangeably which can be confusing as a reader. Authors are suggested to use uniform wording throughout the text.

6. Authors are also suggested to include in table 1 formal test to assess over-representation of any of the groups; i.e., if the study population is over-represented by one of the categories of the demographic variables for eg: it seems females are over represented in the sample or whether participants from a particular education category/age group were more likely to participate than others.

7. Authors present correlations in suppl table 1 between SPQ and depression and anxiety. It will be equally useful to present the correlations between all three outcomes; i.e., also between depression and anxiety.

8. The difference between suppl. table 2 and Table 3 is not clear to me. Aren't they both the model fit indices for predictor models separated by timepoint?

9. Authors are suggested to mention the confounding variables as footnote for each table.

10. The Results section in the abstract is not consistent with results included in the main body of the manuscript. For e.g.: Abstract states "The results revealed that COVID-19-related life concerns were significantly associated with schizotypy in the autumn 2020 and spring 2021 surveys, and with anxiety and depressive symptoms in all surveys"; however, the COVID-19-related life concerns are associated with SPQ in all survey but spring 2020. Similarly, the results have been incorrectly stated for social diversity too. Finally it is just in winter 2021 that the authors observe the mediating effect of excessive media use and not "several surveys" as stated by the authors in abstract.

---

## [Author Response]

Essential revisions:1) Methodologically, I would suggest conducting a sensitivity analysis. Pool the data from all four time points together. You can use generalized equation modeling (GEE) to address repeated measures. To examine the association between the time of the survey and the outcome variables, the date of the survey can be a categorical variable (with April/May 2020 being the reference category) in the model. It is a more elegant way of presenting the results and enables you to statistically compare the outcomes across different periods during the pandemic (using marginal effect).

Thank you for these suggestions.

We have conducted a sensitivity analysis, excluding those 27 individuals. The model fit remains stable, only changes in the third digit after the point are detected (please compare to Table 5.1.). We are presenting this table in the supplementary materials (Supplementary Table 3), and referring to the analysis in the main text.

“The models were calculated with all participants. We then performed a sensitivity analysis calculating the same models without the 27 subjects who participated at all four time points. As a result, the model fit did not change (see Suppl. Table 4).”

Furthermore, we conducted pooled the data as suggested and used survey time-point as a categorical predictor variable in a GEE for our outcome variables. Results match the findings reported in the first three lines of Table 2 in the manuscript.

We have added the results of the GEE in a Suppl. Table 2. and refer to it in the main text:

“As a control analysis we conducted Generalised Estimate Equation model (Carey et al., 2022). Data from all four surveys were pooled, with the first survey acting as the reference for our three outcome variables; the results match those of the Kruskal-Wallis test (Supplementary Table 2).”

2) Correlation value (r) of less than 0.7 – even if the p-value is small (< 0.05) – does not mean a strong correlation between two variables. Please correct the reference on Page 16-17 of the manuscript.

Thank you for pointing out this inaccuracy. We have adjusted the interpretations of the correlation coefficients throughout the manuscript.

“Pearson's correlation coefficient indicated significant moderate correlations between SPQ and anxiety (r=0.57- r=0.68) and between SPQ and depression (r=0.53 – r=0.58) in all four surveys (Supplementary table 3).”

3) Please address some of the discrepancies noted by reviewer #3 [i.e., reference to the Supplementary files, timing of the survey (months vs. season)].

We corrected and clarified all points accordingly: We have (1) updated the numbering to match the order of appearance, (2) corrected all references to supplementary tables and analyses are clearly referred to using e.g. “Supplementary Table 1”, and (3) replaced seasons with the months of survey conduction.

4) Expand the limitation section of the paper to discuss: (a) lack of generalizability; and (b) use of social media for recruitment.

Thank you for this note. We added the following paragraph in the discussion.

“Second, the sample might be biased due to the recruitment strategy. Recruitment was performed through print and social media; however, the questionnaire was only available online, so people without internet access or less ability in using the internet were either excluded from participation or had to rely on people guiding them through the survey. This especially applies to older individuals, who have less access to the internet than young people (Prescott, 2021; Quittschalle et al., 2020). However, we were able to recruit individuals from ages 18-93, with 16.73% aged above 60, which shows a good general representation of age. Since excessive media consumption was investigated in this study, the recruitment strategy, especially via the Internet, may have led to a bias, reaching a disproportionately large number of people who use media excessively. However, in the first investigation (Knolle et al., 2021) using the same recruitment strategy we found that media consumption increased during the COVID-19 pandemic compared to prior to the pandemic. Also, a sampling bias may have occurred over-representing individuals attracted to the topic of mental health (Andrade, 2020), which could one the one hand heighten the strength of the observed associations compared with a representative general population sample and on the other explain the overrepresentation of individuals with higher educational backgrounds. Biases like these are difficult to overcome especially in psychological research. These limitations may affect the generalizability and representativeness of the study.”

(5) Format the Supplementary file to be user-friendly (i.e., put them in word or excel tables).

We have formatted the tables, made them clearer and, as Reviewer #2 suggested, numbered them consecutively. We hope the presentation is now more user-friendly and easier to understand.

Reviewer #1 (Recommendations for the authors):I read the manuscript with interest as the topic was exciting. The manuscript is comprehensive in all sections. However, some comments need to be considered prior to any decision. My comments are as follows:

Thank you for your interest in our work. We have considered your comments in our manuscript. See below for a point by point response.

1. Title: Reflecting the sample size in two countries in the title of the manuscript could better show its scope.

Thank you for your comment. We have adjusted the title to now also reflect that the data has been collected mainly in Germany and the UK. We feel, however, that stating the varying sample sizes for each sample would make the title too complex and long. Please, see our suggestion for the title.

“Investigating the relationship of COVID-19 related stress and media consumption with schizotypy, depression and anxiety in cross-sectional surveys repeated throughout the pandemic in Germany and the UK”

2. Abstract: The methods part of the abstract is too short and needs to be expanded for a better understanding of the results part in the abstract. For example, the criteria or definition of the included conditions of schizotypy, depression, and anxiety.

We have added the questionnaires used to assess the constructs of schizotypy, anxiety, and depression in the methods section of the abstract.

“We assessed schizotypy, depression and anxiety as well as, healthy and unhealthy behaviours and a wide range of sociodemographic scores using online surveys from residents of Germany and the United Kingdom over one year during the COVID-19 pandemic. Four independent samples were collected (April/ May 2020: N=781, September/ October 2020: N=498, January/ February 2021: N=544, May 2021: N= 486). The degree of schizotypy was measured using the Schizotypal Personality Questionnaire (SPQ), anxiety and depression symptoms were surveyed with the Symptom Checklist (SCL^-^27), and healthy and unhealthy behaviors were assessed with the CoronaVirus Health Impact survey (CRISIS). Structural equation models were used to consider the influence of COVID-19 related concerns and social adversity on depressive and anxiety-related symptoms and schizotypal traits in relation to certain healthy (sleep and exercise) and unhealthy behaviours (alcohol and drug consumption, excessive media use).”

3. Introduction: This section is very well prepared and introduces the issue in an adequate manner.

Thank you for your positive feedback.

4. Methods: Although the authors referred to a publication regarding the questionnaire, it is highly suggested that authors briefly introduce the process of preparation and validation of this questionnaire for two countries of Germany and the UK, in the first section of the methods.

We have expanded the description of the questionnaires and presented them in more detail.

“The self-report online survey was conducted using three standardized questionnaires: the Symptom Checklist 27, the Schizotypal Personality Questionnaire (Raine, 1991), the CRISIS survey, and additional demographic information (age, self-reported gender, education and parental education, living conditions). The Coronavirus Health Impact Survey (CRISIS, http://www.crisissurvey.org/) (Nikolaidis et al., 2021) was used to assessed COVID-19 exposure (infection status, symptoms, contact), subjective mental and physical health, and healthy and unhealthy behaviour (i.e., weekly amount of sleep and exercise, consumption of alcohol, drugs, and media). The survey was created by Merikangas, Milham and Stringaris (2020) in the wake of the COVID-19 crisis in order to enable researchers and care providers to examine the extent and impact of life changes induced by the epidemic on mental health and behaviours of individuals and families across diverse international settings. A recent study using a large pilot sample from the US and UK with the CRISIS found that the results are highly reproducible and indicate a high degree of consistency in the predictive power of these factors for mental health during the COVID-19 pandemic (Nikolaidis et al., 2021).We assessed the general mental health status using the short form of the Symptom Check List (SCL^-^27) (Hardt et al., 2011; Hardt and Gerbershagen, 2001; Kuhl et al., 2010). The SCL’s 27 items assess symptoms of anxiety, depression, mistrust, and somatisation. It is a commonly used, economical, multidimensional screening instrument with good psychometric properties (Hardt et al., 2011; Hardt and Braehler, 2007; Kuhl et al., 2010). Finally, we assessed schizotypy using the of the Schizotypy Personality Questionnaire (SPQ, dichotomous version). It is a useful and widely used screening for schizotypal personality disorder in the general population (Raine, 1991).”

5. Methods: The used term "robust ANOVAS" in the statistical analysis section is somehow vague and needs further clarification.

We have made the description of the method of robust ANOVA more explicit.

P 11: “To further explore the differences between the countries and time points in CRISIS variables we conducted robust ANOVAS (Mair and Wilcox, 2020) using the R package WRS2 with country (UK, Germany) and survey (April/ May 2020, September/ October 2020, January/ February 2021, May 2021) as a between-subjects factor. This method was chosen because the data did not meet the parametric assumptions of ANOVA (normality, equal variance, outliers) and the robust ANOVA version is designed for non-parametric data using a maximum-likelihood estimator and therefore less sensitive to non-normality, unequal variance, and outliers compared to normal ANOVA.”

6. Results, figures, and tables: Very well prepared and summarized.

Thanks a lot for this positive response.

7. Discussion: One paragraph is needed on the public health implications of the conducted investigation and expansion on the practical measures for policy-makers and health authorities.

Thank you for this very important comment. We have added the following paragraph to this point:

“Important implications can be drawn from the results for possible future pandemics and national or global health emergencies in order to mitigate a drastic deterioration of the mental health of the general population. To prevent severe isolation and loneliness, social contacts should be allowed in secure and low risk settings, such as meeting outside in pairs wearing masks and, if necessary, keeping some distance. In general, People who live alone are particularly affected by loneliness, which is a major risk factor for experiencing many negative mental states, including psychosis-like experiences (Butter et al., 2017; Le et al., 2019). A recent study has shown that anxiety can be strongly reduced by half by the mere presence of another person (Qi et al., 2020). Furthermore, as excessive media consumption exacerbates the negative effects of COVID-19 related stress and social adversity, it is important to educate individuals on healthy and responsible media consumption. It is necessary to implement these educational measures in school curriculums and professional training settings. During the pandemic we saw an increase in mental health crises, and symptoms of depression and anxiety. Healthcare providers or city councils could offer free online sessions to foster resilience and mental well-being, such as mindfulness, which has been shown to improve general mental health in adults (Antesberger et al., 2021). Critically, special attention should be paid to people with pre-existing mental health conditions or at high risk of developing disorders such as psychosis, as such individuals are more likely to experience exacerbating symptoms during challenging circumstances such as the COVID-19 pandemic (Fekih-Romdhane et al., 2021; Iob et al., 2020; Preti et al., 2020). Further expansion of online psychiatric interventions could be beneficial, as these have shown promise in ameliorating symptoms (Cheli et al., 2020; DeLuca et al., 2020; Schleider et al., 2022).”

Reviewer #2 (Recommendations for the authors):1. I am not certain if it's okay to have ethics approval from only one participating country (Germany) for participants enrolled in the UK and Germany.

The ethical approval was provided for data collection in the UK and Germany; all data security measures are met and is stored exclusively on a server of the Technical University of Munich, in Germany.

2. Though the authors mention their strategy of recruitment for follow-up surveys, it is unclear whether the same recruitment strategy was followed for first survey too.

Thank you very much for pointing this out. We have indeed presented this somewhat inaccurately. Now it should be more clearly understandable.

“The subjects were contacted at the first time point via print media (i.e. Sueddeutsche Zeitung) and social media (i.e. Facebook, Twitter, WhatsApp) and asked to forward the questionnaire to friends and family according to the snowball sampling strategy. At the following sampling time points, respondents who had provided their email address were contacted again and additional participants were recruited via social media and recruitment platforms (www.callforparticipants.com).”

3. It is not clear if the 26 participants which participated in all 4 surveys were included or excluded from the analysis. If included, authors are suggested to perform sensitivity analysis after excluding these participants.

Please excuse the inaccuracy. It was 27 (not 26 as originally stated) individuals who participated in all 4 surveys. In the original analysis those individuals are included.

We have conducted a sensitivity analysis, excluding those 27 individuals. The model fit remains stable, only changes in the third digit after the point are detected (please compare to Table 5.1.). We are presenting this table in the supplementary materials (Supplementary Table 3), and referring to the analysis in the main text.

“The models were calculated with all participants. We then performed a sensitivity analysis calculating the same models without the 27 subjects who participated at all four time points. The models fit equivalently to the full sample models and findings remained unchanged (see Suppl. Table 4).”

4. Supplementary tables are not correctly cited in the text. For e.g.: there is no table in supplementary material titled as supplementary table 4 but is cited in the main text. Also it is confusing to interpret what authors mean by Table X in supplementary material. Authors are suggested not to use titles such as "Table 4" and "Table 2" in supplementary material instead prefix them with word "supplementary". At two instances authors have vaguely mentioned supplementary materials in section 3.3, authors are suggested to cite the supplementary table no. they are referring to. Finally supplementary tables should appear in the order citied in the text (for example suppl table 2 appears before suppl table 1 in text).

Thank you for this suggestion. We have (1) updated the numbering to match the order of appearance, and (2) all references to supplementary tables and analyses are clearly referred to using e.g. “Supplementary Table 1”.

5. Authors use some words interchangeably, for example 1: Though authors clarify the usage of gender (and not biological sex), this has been interchangeably mentioned in section 2.4, pg 13. Example 2: Authors use the words may 2021 and spring 2021 interchangeably which can be confusing as a reader. Authors are suggested to use uniform wording throughout the text.

We have replaced seasons with the exact month throughout the text and "sex" with "gender".

6. Authors are also suggested to include in table 1 formal test to assess over-representation of any of the groups; i.e., if the study population is over-represented by one of the categories of the demographic variables for eg: it seems females are over represented in the sample or whether participants from a particular education category/age group were more likely to participate than others.

Thank you. Please see, Table 1 which assesses the distributions of certain group characteristics (six columns on the right). The samples did not differ in terms of gender, but they did differ in terms of education level, area of residence, and prior mental health status. We have discussed this point and especially the resulting problem of generalizability in the limitations section.

7. Authors present correlations in suppl table 1 between SPQ and depression and anxiety. It will be equally useful to present the correlations between all three outcomes; i.e., also between depression and anxiety.

Thank you for this suggestion. We added a new column with Pearson correlation coefficients between anxiety and depression scores (Supplementary Table 3).

8. The difference between suppl. table 2 and Table 3 is not clear to me. Aren't they both the model fit indices for predictor models separated by timepoint?

No, this is not the same content (compare title). In suppl. table 2 (now Table 1, supplements) the model fit indices of the two prediction models COVID-19 related life concerns (Financial Impact + Restrictions perceived as stressful + concerns about life stability) and social adversity (loneliness + stressful social relationship changes + negative thoughts) are presented. In suppl. table 3 (now Table 5.1 – 5.8, supplements) the complete models including mediators (Drugs, Media, Alcohol, Sleep, Exercise) and outcome variables (Anxiety, SPQ score, Depression) are presented.

9. Authors are suggested to mention the confounding variables as footnote for each table.

Thank you for this suggestion. We added the sentence: “Mediators and outcome variables were controlled for the confounding variables of country of residence (Germany vs. UK), age, gender, highest level of education, living conditions, and health status before COVID 19” in the note lines in Tables 4, 5, 6.

10. The Results section in the abstract is not consistent with results included in the main body of the manuscript. For e.g.: Abstract states "The results revealed that COVID-19-related life concerns were significantly associated with schizotypy in the autumn 2020 and spring 2021 surveys, and with anxiety and depressive symptoms in all surveys"; however, the COVID-19-related life concerns are associated with SPQ in all survey but spring 2020. Similarly, the results have been incorrectly stated for social diversity too. Finally it is just in winter 2021 that the authors observe the mediating effect of excessive media use and not "several surveys" as stated by the authors in abstract.

Thank you very much for noticing this. We have compared the result parts again and adjusted them accordingly in the abstract.

“The results revealed that COVID-19 related life concerns were significantly associated with schizotypy in the September/ October 2020 and May 2021 surveys, with anxiety in September/ October 2020, January/ February 2021, and May 2021 surveys, and with depressive symptoms in all surveys. Social adversity significantly affected the expression of schizotypal traits and depressive and anxiety symptoms in all four surveys. Importantly, we found that excessive media consumption (>4h per day) fully mediated the relationship of COVID-19 related life concerns and schizotypal traits in the January/ February 2021 survey. Furthermore, several of the surveys showed that excessive media consumption was associated with increased depressive and anxiety-related symptoms in people burdened by COVID-19 related life.”